# Beyond Misfolding: A New Paradigm for the Relationship Between Protein Folding and Aggregation

**DOI:** 10.3390/ijms26010053

**Published:** 2024-12-24

**Authors:** Seong Il Choi, Yoontae Jin, Yura Choi, Baik L. Seong

**Affiliations:** 1Department of Pediatrics, Severance Hospital, Institute of Allergy, Brain Korea 21 PLUS Project for Medical Science, Yonsei University College of Medicine, Seoul 03722, Republic of Korea; 2Vaccine Innovative Technology ALliance (VITAL)-Korea, Seoul 03722, Republic of Korea; yoontae_jin@yonsei.ac.kr (Y.J.); yurachoi@yonsei.ac.kr (Y.C.); 3Department of Microbiology and Immunology, Institute for Immunology and Immunological Diseases, Graduate School of Medical Science, Brain Korea 21 Project, Yonsei University College of Medicine, Seoul 03722, Republic of Korea; 4Department of Integrative Biotechnology, Yonsei University, Incheon 21983, Republic of Korea; 5Department of Microbiology, College of Medicine, Yonsei University, Seoul 03722, Republic of Korea

**Keywords:** protein folding, misfolding, aggregation, charges, molecular chaperones, Anfinsen’s thermodynamic hypothesis, excluded volumes, intermolecular repulsive forces, proteome solubility, metastability, proteinopathies

## Abstract

Aggregation is intricately linked to protein folding, necessitating a precise understanding of their relationship. Traditionally, aggregation has been viewed primarily as a sequential consequence of protein folding and misfolding. However, this conventional paradigm is inherently incomplete and can be deeply misleading. Remarkably, it fails to adequately explain how intrinsic and extrinsic factors, such as charges and cellular macromolecules, prevent intermolecular aggregation independently of intramolecular protein folding and structure. The pervasive inconsistencies between protein folding and aggregation call for a new framework. In all combined reactions of molecules, both intramolecular and intermolecular rate (or equilibrium) constants are mutually independent; accordingly, intrinsic and extrinsic factors independently affect both rate constants. This universal principle, when applied to protein folding and aggregation, indicates that they should be treated as two independent yet interconnected processes. Based on this principle, a new framework provides groundbreaking insights into misfolding, Anfinsen’s thermodynamic hypothesis, molecular chaperones, intrinsic chaperone-like activities of cellular macromolecules, intermolecular repulsive force-driven aggregation inhibition, proteome solubility maintenance, and proteinopathies. Consequently, this paradigm shift not only refines our current understanding but also offers a more comprehensive view of how aggregation is coupled to protein folding in the complex cellular milieu.

## 1. Introduction

Proteins need to correctly and efficiently fold while additionally maintaining their solubility against aggregation to perform their biological functions in the crowded cellular milieu [1,2,3]. During their lifespan, proteins undergo kinetic and thermodynamic partitioning between folding and aggregation [4,5,6,7]. Newly synthesized proteins can aggregate even before folding, compromising the yield of productive protein folding [8,9,10]. Molecular chaperones are known to assist in protein folding by preventing misfolding and aggregation, and they play a crucial role in protein homeostasis (proteostasis) and proteinopathies [1,10,11,12,13,14]. The traditional studies of protein folding are conducted in the absence of aggregation. However, it is increasingly essential to understand cellular protein folding in the context of aggregation [9,10,11,15]. Aggregation is associated with impaired proteostasis and proteinopathies, including Alzheimer’s disease (AD), Parkinson’s disease (PD), amyotrophic lateral sclerosis (ALS), and prion diseases [16]. These aggregation-associated diseases are also referred to as protein folding or misfolding diseases [17,18,19]. According to Anfinsen’s thermodynamic hypothesis, the native three-dimensional structure of a protein is supposed to be thermodynamically the most stable under physiological conditions, and thus, it is encoded in its amino acid sequence [20]. However, aggregates such as amyloid fibrils have been proposed to be thermodynamically more stable than their folded, functional structures in the combined landscapes of protein folding and aggregation [6,21,22,23]. This metastability challenges Anfinsen’s thermodynamic hypothesis [6,21,23,24]. The metastability issue extends beyond protein folding and Anfinsen’s thermodynamic hypothesis. Thus, it remains a grand challenge to understand how the cellular proteome kinetically and thermodynamically maintains solubility at the risk of aggregation [25]. Given its significant impact on protein folding, proteostasis, and proteinopathies, aggregation remains a central yet unsolved issue in both scientific and medical research.

Traditionally, aggregation has been viewed either as a sequential consequence of protein folding (or misfolding) or as a part of a continuum with protein folding [1,2,6,15,16,17,18,19,21,22,23,24,25]; thus, aggregation is perceived as being strictly dependent on protein folding and misfolding. This widely accepted paradigm has solidified into a dogmatic framework that shapes understanding of the relationship between protein folding and aggregation across various academic disciplines. The adherence to this paradigm is primarily due to the following reasons. Firstly, proper protein folding and high thermodynamic stability are decisive factors in preventing aggregation. Secondly, misfolded structures can act as direct precursors for aggregation, supporting a sequential pathway from folding to misfolding to aggregation. Thirdly, the physicochemical properties of protein monomers, including conformation, charges, and hydrophobicity, are crucial for their aggregation behaviors [26,27]. Finally, the folding and aggregation landscapes of proteins with identical amino acid sequences can be understood as continuous conformational navigation [28,29,30]. So far, the limitations, pitfalls, and misconceptions of the conventional paradigm have not been widely recognized or addressed in the protein science community. This oversight can be attributed to the well-established foundations of the conventional paradigm, its broad acceptance in the field, and the dominant approaches of characterizing diverse protein conformational landscapes through protein folding, structure, and conformational change. In contrast, while conventional protein folding (or misfolding) is the intramolecular conformational change within a single protein molecule, aggregation is an intermolecular association between multiple aggregation monomers, dependent on protein concentration. Despite their interconnection, intramolecular and intermolecular reactions, such as protein folding and aggregation, exhibit fundamental differences. Both processes are governed by their distinct rate constants, intramolecular for protein folding and intermolecular for aggregation, each operating independently; consequently, protein folding and aggregation should be regarded as two independent yet interconnected processes rather than sequential ones [3,31,32].

The independence between protein folding and aggregation, while initially counterintuitive, is supported by compelling evidence. For instance, charges play a crucial role in governing the solubility of molecules against aggregation, including small chemicals, peptides, cellular macromolecules, and colloids, by generating intermolecular repulsions, including desolvation penalties [33,34,35,36,37]. Charges of proteins play a key role in preventing intermolecular aggregation while having minimal or adverse effects on the intramolecular folding rate and thermodynamic stability of proteins [38]. Therefore, charges are often referred to as ’aggregation gatekeepers’ [38,39]. Consistent with this designation, it is difficult or impossible to explain the intermolecular repulsion-mediated aggregation inhibition with their effects on protein folding and protein structure [3,32], illustrating a significant limitation of the conventional paradigm and modern protein science. Contrarily, the influence of charges can be described by their independent effects on both intramolecular protein folding and intermolecular aggregation [32]. A significant fraction of the proteome consists of intrinsically disordered proteins (IDPs) and intrinsically disordered regions (IDRs) [40,41]. Their solubility maintenance against aggregation is difficult to explain through classical protein folding and thermodynamic stability. Instead, their solubility is primarily determined by the charged or hydrophilic residues of their amino acid sequences [42]. Cellular macromolecules with numerous surface charges and large excluded volumes are proposed to exhibit generic intrinsic chaperone-like activities in a manner similar to small-charged residues of proteins as aggregation gatekeepers [43,44,45]. Similarly, the prevention of intermolecular aggregation by molecular chaperones is challenging to explain with protein folding [3,32]. Traditionally, the effects of cellular macromolecules on aggregation have been largely interpreted or assumed based on their effects on protein folding, which is consistent with the conventional paradigm. However, given the independence between protein folding and aggregation, all intrinsic and extrinsic factors independently affect intramolecular protein folding and intermolecular aggregation [3,31,32]. This independence necessitates a redefinition of how the cellular milieu, including cellular macromolecules, influences aggregation, as elaborated throughout this paper.

A new framework, founded on the principle of rate constant independence, delineates the relationship between protein folding and aggregation as two independent yet interconnected processes. It provides clear insights into previously controversial, unsolved, and unrecognized issues at the interface between protein folding and aggregation, especially within the complex cellular milieu. As illustrated in Figure 1, these issues include misfolding, Anfinsen’s thermodynamic hypothesis, molecular chaperones, intrinsic chaperone-like activities of cellular macromolecules, pitfalls of modern protein science, intermolecular repulsive force-driven aggregation inhibition, metastability issue in vivo, and proteinopathies.

## 2. New Framework for the Relationship Between Protein Folding and Aggregation

This section introduces a new framework that redefines the relationship between protein folding and aggregation, positing that these processes are independent yet interconnected [31]. Proteins navigate diverse conformational landscapes, including protein folding, quaternary structure formation, and native interactions or complexations with cellular macromolecules and ligands. Additionally, proteins undergo various forms of aggregation, such as oligomers, amorphous aggregates, ordered aggregates, and liquid–liquid phase separation. The minimal aggregates are aberrant dimers [46]. Establishing the relationships between these diverse protein conformational landscapes is a prerequisite for understanding their interplay.

Each conformational landscape that proteins navigate is kinetically and thermodynamically shaped by its own set of forward and reverse rate (or equilibrium) constants at both microscopic and macroscopic levels, analogous to general chemical reactions [31,32]. These rate constants enable theoretical derivations of overall reaction equations, allowing quantitative predictions over time for reactant, product, and intermediate concentrations. Crucially, each rate constant is mutually independent in all physical and chemical reactions without exception. Therefore, the standard protein folding and aggregation free energy landscapes, defined by their respective rate constants, are independent, as illustrated in Figure 2. Consequently, all intrinsic and extrinsic factors influence these landscapes independently. Notably, two independent landscapes are interconnected or communicated through shared free aggregation monomers [31]. These monomers can exist in various states, including unfolded, misfolded, partially folded, or fully folded forms. Throughout this paper, unfolded structures are used as representative shared aggregation monomers unless otherwise stated. Consistently, protein folding and aberrant aggregation are interconnected through unfolded, misfolded, or folding intermediate structures [23,47,48,49,50,51]. Importantly, the aggregation landscapes are shaped by the interplay between shared aggregation monomers and their intermolecular assembly into aggregates. At a given total protein concentration, the protein folding free energy landscape determines the concentration of shared aggregation monomers, ‘potentially’ influencing aggregation, in line with the conventional paradigm. In contrast to the conventional paradigm, however, the ‘actual’ intermolecular aggregation is governed by the aggregation free energy landscape, operating independently of the protein folding free energy landscape [3,31,32]. Therefore, at a given total protein concentration, the final amount of aggregation or protein solubility against aggregation, as well as productive protein folding yield in the presence of aggregation, is determined by both intramolecular and intermolecular rate constants. This new framework treats the combined protein folding and aggregation in a similar way to how combined intramolecular and intermolecular reactions in non-protein molecules are generally understood.

There are several key differences between the new framework and the conventional paradigm. Firstly, the new framework not only encompasses but also extends the conventional paradigm, providing a more accurate linkage between protein folding and aggregation. Of note, real analyses of combined protein folding and aggregation utilize rate constants to describe these combined processes [52,53,54]. Secondly, the new framework reveals that intrinsic and extrinsic factors independently influence the landscapes of protein folding and aggregation. Thirdly, the conventional paradigm primarily focuses on the forward reactions from folding to misfolding to aggregation, which fails to adequately address intermolecular aggregation prevention and reverse reactions, such as disaggregation from aggregates back to soluble monomers. This limitation becomes apparent when trying to explain how charges and cellular macromolecules, including molecular chaperones, inhibit aggregation independently of protein folding, structure, and conformational change [3,31,32]. In contrast, these conundrums can be easily resolved by the new framework that accounts for both forward and reverse rate constants and the independent effects of various factors on protein folding and aggregation. Finally, the new framework redefines shared aggregation monomers not just as temporary intermediates but as main reactants in aggregation processes. This redefinition is essential for addressing the relationship between Anfinsen’s thermodynamic hypothesis and aggregation, as well as the issue of metastability. Moreover, these shared aggregation monomers can encompass any intramolecular structures, thereby broadening the applicability of the new framework to explain various types of aggregation. Furthermore, the new framework for describing aggregation aligns with that of non-protein molecules, bridging the understanding gaps between protein science and material science, such as colloidal science, as discussed in Section 7.1.

The pervasive inconsistencies between protein folding and aggregation, which challenge the conventional paradigm, stem from their independence and the independent effects of various factors on each process. Solubility maintenance against aggregation can be achieved even at the expense of protein folding rate and thermodynamic stability, as the actual aggregation of shared monomers is governed by the intermolecular rate constants shaping the aggregation free energy landscape. For instance, the role of charges as the aggregation gatekeepers and the solubility maintenance of IDPs can be elucidated by the new framework. Both protein folding and aggregation are molecular compactions. Thus, factors can stabilize (or destabilize) both processes in a similar manner, thereby exacerbating these inconsistencies. For instance, hydrophobic interactions are the major stabilizers for both protein folding and aggregation; thus, protein stabilization through increased hydrophobic interactions can paradoxically enhance the stability of aggregation [39]. Molecular chaperones, including small heat shock proteins, bind to the exposed hydrophobic regions of clients [10,55,56]. Therefore, they can inhibit both protein folding and aggregation. Supercharging inhibits aggregation while also thermodynamically destabilizing proteins [57]. The net charges of proteins are influenced by environmental pH. Proteins denatured at extreme pH can paradoxically resist amyloid fibril formation due to intermolecular electrostatic repulsions [58]. Conformational entropy destabilizes the molecular compaction of both protein folding and aggregation [31]. The excluded volume effect by macromolecular crowding stabilizes more compact structures, such as folded structures and aggregates [59]. The new framework can qualitatively and quantitatively explain these pervasive inconsistencies by combining the independent effects of intrinsic and extrinsic factors on intramolecular protein folding and intermolecular aggregation.

## 3. Relationship Between Misfolding and Aggregation

Misfolding is commonly linked with aggregation, supporting the conventional paradigm. According to the Lumry–Eyring model, irreversible misfolding is considered a critical step for aggregation [60,61]. In funnel-shaped protein folding energy landscapes, conformers trapped in local free energy minima are prone to aggregation [62,63,64]. Molecular chaperones perform a quality control function by recognizing and rescuing the non-native conformers of their clients [1,10]. Consequently, it is widely believed that aggregation results from misfolding, often leading to a lack of distinction between the two processes. However, the relationship between misfolding and aggregation is analogous to that between protein folding and aggregation. Therefore, aggregation extends beyond misfolding. Importantly, the prevention of misfolding by various factors is independent of their prevention of intermolecular aggregation among misfolded monomers [32]. This independence is important for understanding the mechanisms of action of molecular chaperones and therapeutic drug candidates targeting proteinopathies. Designing drugs to prevent misfolding or to prevent aggregation involves distinct approaches in drug discovery for proteinopathies, underscoring the need for a rigorous distinction between misfolding and aggregation.

Contrary to the conventional paradigm, aggregation does not necessarily result from misfolding. The shared aggregation monomers depicted in Figure 2 can exist as unfolded, misfolded, partially folded, or fully folded structures. For instance, the crystallization of folded proteins and the formation of actin fibers occur independently of misfolding. It is important to note that unfolded structures are distinct from misfolded structures. To address the diversity of aggregation monomers, it is useful to differentiate between misfolding, unfolding, and aberrant aggregation, as illustrated in Figure 3. According to the energy landscape theory of protein folding, misfolding occurs when proteins adopt stable non-native conformations that become kinetically trapped in local minima near the folded state, impeding proper folding [62]. This definition indicates that non-native structures found in aggregates are not necessarily misfolded structures. Additionally, the prevention of misfolding can be experimentally supported by the increased folding rates observed in the absence of aggregation [65,66]. Molecular chaperones can increase the folding rates of clients by lowering kinetic barriers, smoothing navigational landscapes, or employing an iterative annealing mechanism that involves melting and refolding misfolded structures [66,67]. In contrast to misfolding, unfolding involves a transition to higher energy unfolded structures, which are generally not considered to be kinetically trapped in the intramolecular conformational navigation. Moreover, conversions between these unfolded ensembles occur rapidly. The aggregation of unfolded structures, such as IDPs, IDRs, and unfoldable peptides, is difficult to explain solely through misfolding. Similarly, small domain-sized proteins often fold in a two-state manner (folded vs. unfolded states) [68]. The two-state protein folding is considered an effective strategy to avoid misfolding [69], illustrating the distinction between unfolding and misfolding. For instance, the short peptides of seven amino acid residues (e.g., GNNQQNY) make the stable amyloid fibers [70,71]. These unfoldable peptides are expected not to have a stable kinetic trap or misfolded state at the intramolecular level. From the viewpoint of intramolecular conformational changes, aggregates behave like deep stable traps; however, it is important to recognize that aggregates are intermolecular assemblies, distinct from intramolecular misfolding and unfolding, independent of protein concentration. Reversible small aggregates can be easily mistaken for folding intermediates or kinetically trapped misfolded structures in protein folding studies [72]. Moreover, the role of molecular chaperones in preventing reversible aggregates can be misinterpreted as an enhancement of the folding rate of client proteins [73]. Although misfolding closely correlates with aggregation, it is essential to recognize that misfolding is not the sole cause of the aggregation of non-native structures.

## 4. Relationship Between Anfinsen’s Thermodynamic Hypothesis and Aggregation

Anfinsen’s thermodynamic hypothesis is a cornerstone in the study of protein folding. Aligned with the conventional paradigm, it is often believed to encompass protein aggregation. For instance, the amino acid sequence encoding native structures also encodes more thermodynamically stable aggregates such as amyloid fibrils [74]. Historically, this has led to the proposal that aggregates, such as prions and amyloids, challenge Anfinsen’s thermodynamic hypothesis [75]. However, recognizing the independence between protein folding and aggregation provides clarity on how Anfinsen’s thermodynamic hypothesis relates to aggregation, as illustrated in Figure 4.

Anfinsen’s thermodynamic hypothesis is strictly limited to intramolecular conformational changes and their relative stability. This hypothesis does not explicitly address the concept of aggregation [22,31]. In Anfinsen’s thermodynamic hypothesis, the standard Gibbs energy for protein folding is expressed as Δ$G_{N-D}^{0}$ = −*RT* ln *K*, where *K* = [N]/[D] is the equilibrium constant for protein folding determined by the ratio of folded (N) to unfolded (D) states at equilibrium. These values are generally obtained by the equilibrium unfolding experiments [76]. In contrast, the standard Gibbs energy for converting soluble aggregation monomers (D) into fibrils (A) can be given by Δ$G_{A-D}^{0}$ = −*RT* ln *K*_a_ = *RT* ln $M_{s}^{max}$, where *K*_a_ and $M_{s}^{max}$ are the equilibrium constant for elongation and the solubility limit of monomers, respectively [6,7,77]. The standard Gibbs energies for protein folding and aggregation, calculated from their respective equilibrium constants under standard conditions (where reactants and products are fixed at 1 mole each), are independent. The concentration of these aggregation monomers is determined by the following equation: D = *C*_total_/(*K* + 1), where *C*_total_ represents the given initial total protein concentration of soluble monomers, and *K* is the equilibrium constant for protein folding [78]. This equation shows that the intramolecular protein folding thermodynamics affects aggregation by determining the concentration of these unfolded monomers, which act as the main reactants for aggregation, in line with the new framework.

Metastability in protein aggregation is traditionally understood by conceptually comparing the thermodynamic stability difference (Δ$G_{A-N}^{0}$) between native folded structures (N) and aggregates such as amyloid fibrils (A) [21,25]. This topic will be explored in more detail later in Section 8. However, direct calculation of Δ$G_{A-N}^{0}$ in equilibrium thermodynamics poses challenges due to the independence between intramolecular thermodynamics and intermolecular thermodynamics. Thus, Δ$G_{A-N}^{0}$ is indirectly obtained by combining Δ$G_{D-N}^{0}$ with Δ$G_{A-D}^{0}$ (Δ$G_{A-N}^{0}$ = Δ$G_{D-N}^{0}$+ Δ$G_{A-D}^{0}$), as depicted in Figure 4. Direct thermodynamic comparison (Δ$G_{A-N}^{0}$) is feasible when the aggregation monomers in aggregates are folded structures, such as actin fibers and crystalline forms of native proteins. However, for unfoldable polypeptides, such as IDPs and IDRs, where the unfolding step (Δ$G_{D-N}^{0}$) is absent, the aggregation thermodynamic stability relies solely on Δ$G_{A-D}^{0}$, similar to the aggregation thermodynamics of non-protein molecules. Importantly, this indicates that the metastability issue for unfoldable peptides is explained with Δ$G_{A-D}^{0}$ rather than Δ$G_{A-N}^{0}$. Of note, the aggregation of all molecules, including folded proteins, unfoldable peptides, and non-protein molecules, can be commonly described in terms of Δ$G_{A-D}^{0}$.

According to the independence between Anfinsen’s thermodynamic hypothesis and aggregation, it is evident that aggregation phenomena, including molecular chaperone-assisted protein folding by preventing aggregation, prion and amyloid formation, and the metastability of native states towards aggregates, do not undermine or challenge the validity of Anfinsen’s thermodynamic hypothesis and classical protein folding principles. Due to this independence, however, Anfinsen’s thermodynamic hypothesis is inherently insufficient for fully describing aggregation; aggregation is beyond Anfinsen’s thermodynamic hypothesis. For instance, high thermodynamic stability alone is not enough to guarantee safety against aggregation [79]. The relationship between thermodynamic stability and solubility has long been a controversial issue [79,80], as evident in IDPs as outstanding examples. According to the new framework, a protein’s thermodynamic stability is determined by intramolecular rate constants, whereas its solubility is influenced by both intramolecular and intermolecular rate constants, which are mutually independent. In the refolding experiments conducted by Anfinsen himself, aggregation significantly inhibits the productive refolding yield [81]. Nonetheless, Anfinsen did not argue that his postulate included the concept of aggregation. Protein folding is analogous to chemical isomerization, while aggregation is comparable to crystallization. There is no known instance where the thermodynamics of intermolecular crystallization challenges the intramolecular thermodynamics of isomerization. To integrate Anfinsen’s thermodynamic hypothesis with aggregation, it is essential to recognize that they are two independent realms [32].

## 5. Relationship Between Molecular Chaperones and Aggregation

Molecular chaperones are the most well-known molecules at the interface of protein folding and aggregation. They play a pivotal role in assisting protein folding, primarily by preventing misfolding and aggregation [1,10]. This role challenges the concept of spontaneous protein folding posited by Anfinsen’s thermodynamic hypothesis, suggesting that folding may not always occur spontaneously, independently of external influences [82,83]. Traditionally, the understanding of molecular chaperones’ functions has been centered on their role in assisting protein folding. However, their involvement in aggregation necessitates a re-evaluation of their role through the new framework. The combined protein folding and aggregation in the cellular milieu has been primarily framed by the molecular chaperone concept. Nonetheless, the molecular chaperone concept often overlooks or ignores several features of the cellular milieu that are important for protein folding and aggregation, as detailed in Section 5.3.

### 5.1. Independence Between Direct and Indirect Protein Folding Assistance of Molecular Chaperones

In some instances, molecular chaperones function as foldases, acting as catalysts that increase protein folding rate [65,84,85]. This direct assistance in protein folding is aligned with their role in preventing misfolding. However, more commonly, molecular chaperones decrease folding rates [86,87,88,89,90,91,92,93] and function similarly to chaotropic agents that unfold or disrupt protein structures [94]. Similarly, molecular chaperones are defined as unfolding enzymes [95]. From the traditional protein folding concept, molecular chaperones can generally act as folding inhibitors in the absence of aggregation. The hydrophobic interactions between molecular chaperones, including small heat-shock proteins, and their clients can inhibit both protein folding and aggregation. Therefore, proteins like GroEL/ES, Hsp70, and Hsp90 require an ATP-dependent binding and release cycle to facilitate protein folding [96,97]. Too tight binding of small heat-shock proteins strongly inhibits protein folding. Importantly, molecular chaperones generally increase productive protein folding yield by preventing aggregation at the cost of folding rate [73,88,96,98]. This indirect protein folding assistance by preventing aggregation is independent of the direct protein folding assistance, as illustrated in Figure 5A. This necessitates a rigorous distinction between these independent pathways to avoid a paradox where molecular chaperones seem to assist and inhibit folding simultaneously. This paradox, similar to the inconsistencies between protein folding and aggregation, is resolved by recognizing the independent effects of molecular chaperones on intramolecular protein folding and intermolecular aggregation [3,32]. Debate persists over the primary roles of molecular chaperones as foldases or holdases [99,100,101,102]. More accurately, foldases are associated with intramolecular protein folding, whereas holdases are likely linked to intermolecular aggregation. Therefore, the functions of foldases and holdases should not be considered opposing but rather as independent activities. Thus, molecular chaperones can exhibit both activities. For instance, the trigger factor chaperone of *E. coli* and G-quadruplexes of nucleic acids can act as both foldases and holdases [103,104,105]. In some cases, ATP-dependent molecular chaperones are classified as foldases, while ATP-independent chaperones are categorized as holdases [102,104,106]. However, this classification can be misleading since ATP-dependent molecular chaperones can generally slow down the protein folding rate in the absence of aggregation. In this regard, ATP-independent molecular chaperones, such as trigger factor and Spy that bind clients with weak affinity, likely assist protein folding in a similar manner to ATP-dependent molecular chaperones. Importantly, the new framework highlights the essential role of molecular chaperones in addressing aggregation issues, including their provision of indirect protein folding assistance by preventing aggregation in the combined protein folding and aggregation. These roles, which have been historically less emphasized compared to the direct folding assistance, fundamentally extend beyond (or are challenging to explain with) the traditional concepts of protein folding, misfolding, Anfinsen’s thermodynamic hypothesis, and established protein folding principles.

### 5.2. GroEL/ES Chamber: Delineating Independence Between Protein Folding and Aggregation

The GroEL/ES chamber serves as a compelling model for illustrating the independence between protein folding and aggregation, along with the independent effects of molecular chaperones on both processes [32], as depicted in Figure 5B. Protein folding in the GroEL/ES chamber has been a major focus of molecular chaperone research. Proteins within the GroEL/ES chamber are completely protected from aggregation, allowing correct folding without interference; this protection has led to the chamber being referred to as the ‘Anfinsen cage’ [107]. The steric masking of the GroEL/ES chamber completely prevents the self-aggregation of the client protein in it. Similar to charges acting as aggregation gatekeepers, the intermolecular aggregation inhibition by the GroEL/ES chamber operates independently of the representative cores of modern protein science. These cores include intramolecular protein folding, unfolding, and misfolding, structural stability by confinement, conformational entropy, and the attractive interactions between a protein and the chamber’s wall surfaces. The GroEL/ES chamber, as an extrinsic factor, independently affects protein folding and aggregation. The aggregation landscape in the GroEL/ES chamber is dictated not by the intrinsic properties of client protein but solely by the chamber’s property or its encapsulation [32]. The GroEL/ES chamber, along with charges acting as aggregation gatekeepers, underscores the limitations of the conventional paradigm and modern protein science in accurately describing protein aggregation, especially in vivo. Such limitations will be further elaborated in Section 7.1. Additionally, the GroEL/ES chamber illustrates the independence between foldase and holdase activities; it can modulate protein folding rates—either inhibiting or enhancing them—while independently and simultaneously preventing aggregation.

### 5.3. Overlooked Aspects of the Complex Cellular Milieu in the Molecular Chaperone Concept

Molecular chaperones, essential for protein quality control, are traditionally defined as proteins that assist the folding or assembly of other macromolecules without being part of their final structures [108]. This specific definition can overlook the broader chaperone-like activities present within the complex cellular milieu. Firstly, it excludes the potential chaperone-like activity of non-protein cellular macromolecules, such as nucleic acids (RNAs and DNAs), ribosomes, membranes, polyphosphates, and polysaccharides. Indeed, these non-protein cellular macromolecules have been shown to exhibit chaperone-like activities [103,109,110,111,112,113,114,115]. Secondly, this definition primarily considers the reversible interactions in trans between molecular chaperones and their client proteins, thereby excluding potential chaperone-like activity of cellular macromolecules that are physically connected to polypeptides in cis through covalent or covalent-like linkages. This type of linkage, known as macromolecular tethering, will be further discussed in Section 6.1. Thirdly, the traditional definition also excludes the potential chaperone-like activities of native interactors since molecular chaperones are not components of the final structures of their clients. However, the majority of the cellular proteome is involved in native interactions or complexes with other cellular macromolecules [116,117,118,119,120]. These native interactors or ligands play a crucial role in protein folding and stability [121,122,123,124,125]. Fourthly, the core action mechanisms of molecular chaperones involve hydrophobic interaction-mediated substrate recognition and stabilization against aggregation [1,10]. In contrast, substrate recognition and stabilization through non-hydrophobic interactions, such as attractive electrostatic interactions and intermolecular repulsive forces, respectively, are largely overlooked. Finally, the current concept of proteostasis largely depends on the roles of molecular chaperones in processes such as protein folding, assembly, translocation, and degradation [1,12]. However, this concept may not fully account for the broader array of chaperone-like activities and the critical role of native interactors in protein folding, stability, and aggregation inhibition within the cellular milieu.

## 6. Generic Intrinsic Chaperone-like Activities of Cellular Macromolecules: Their Independent Effects on Protein Folding and Aggregation

The cellular milieu, including cellular macromolecules, has traditionally been understood and described primarily from the perspective of protein folding, leading to a skewed understanding of aggregation. Unlike simplified in vitro environments, the cellular milieu is characterized by macromolecular crowding, where cellular macromolecules significantly occupy cell volume [59,126]. Notably, the excluded volume effect by macromolecular crowding, widely studied in protein science, is based on the assumption that cellular macromolecules act as bystanders, occupying space without directly interacting with proteins. Thus, this effect can allow the principles of protein folding and aggregation observed in vitro to be applied to in vivo conditions. However, a hallmark yet often overlooked feature of the cellular milieu is the physical linkages of polypeptides with cellular macromolecules, which may differ significantly from protein folding and aggregation observed in vitro, governed by the protein’s own intrinsic properties. From their synthesis to degradation, polypeptides are physically linked to a broad spectrum of cellular macromolecules via various types of interactions: covalent or noncovalent, specific or nonspecific, direct or indirect, and transient or stable. These cellular macromolecules include ribosomes, domains in multidomain proteins, lipid bilayers, other proteins, and nucleic acids. Polypeptides engage in transient and nonspecific interactions with crowded cellular macromolecules, known as soft or quinary interactions [127,128]. The representative physical linkages in vivo include interactions with molecular chaperones, macromolecular tethering, and native complexation. The new framework not only reveals that the cellular milieu independently affects protein folding and aggregation, but it can also underlie the intrinsic chaperone-like activities of cellular macromolecules, as described below. Furthermore, it bridges the gaps between combined protein folding and aggregation in vitro and in vivo.

### 6.1. Independent Effects of Macromolecular Tethering on Protein Folding and Aggregation

Macromolecular tethering is a hallmark of de novo protein folding within the cellular milieu. Newly synthesized polypeptides are tightly tethered—via covalent or covalent-like linkages, or in cis—to megadalton-sized ribosomes with supernegative surface charges (entire proteome), other domains in multidomain proteins (80% of the proteome), or gigantic lipid bilayers (30% of the proteome) [129,130,131]. In eucaryotes, proteins undergo modifications such as ubiquitination, sumoylation, and glycosylation [132,133,134]. Thus, macromolecular tethering represents a proteome-wide phenomenon. The new framework highlights that macromolecular tethering independently influences both protein folding and aggregation, as illustrated in Figure 6. Cotranslational folding and cotranslocational folding occur in the context of macromolecular tethering. Historically, the assumption that surface-exposed nascent chains on polysomes are prone to aggregation has justified the necessity of molecular chaperone assistance at ribosomal exit sites, laying the foundation for the widely accepted model of molecular chaperone-assisted de novo protein folding [8,9,10,135,136,137]. This assumption aligns with the conventional paradigm, which posits that if nascent chains exhibit similar folding properties—whether they are free in solution or tethered to ribosomes—their aggregation properties should also be similar. Even though a single charged residue of proteins can significantly influence aggregation inhibition, megadalton-sized ribosomes with thousands of surface charges are treated like ideal gases with no volume or interaction. This interpretation typifies the conventional paradigm’s approach to describing and understanding the cellular milieu in terms of aggregation. Consistently, the concept of cis-linkage is not considered in the traditional molecular chaperone concept. However, evidence from ribosome display technology and the fusion of aggregation-prone proteins indicates that nascent chains tethered to ribosomes are in an aggregation-resistant and folding-competent state [138,139], demonstrating the intrinsic chaperone-like activities of ribosomes in the tethering context. The three-dimensional structures of polysomes show how their arrangement inhibits aggregation by maximizing the distance between nascent chains [140]. In contrast to dramatic effects on aggregation inhibition, the folding pathways of small proteins or domains remain consistent whether they are free in solution or tethered to ribosomes, despite ribosomes destabilizing protein stability [141,142,143]. Taken together, these observations highlight the inconsistencies between ribosome tethering’ effects on protein folding and aggregation. Ribosomal surface charges and large excluded volumes are proposed to protect tethered nascent chains from aggregation, allowing them to fold spontaneously based on their sequence; this is referred to as cis-acting chaperone type [43,45,129]. Understanding de novo protein folding and aggregation—as well as molecular chaperone-assisted de novo protein folding on the ribosome—requires considering ribosome tethering and its independent effects on both processes.

### 6.2. Biological Relevance and Significance of Protein Fusion Approaches

The fusion, or covalent tethering, of soluble-enhancing tags to the amino termini of aggregation-prone proteins is one of the most effective strategies for preventing aggregation and increasing the yield of properly folded proteins in *E. coli* [144]. A variety of fusion tags are available, including maltose-binding protein (MBP), glutathione S-transferase (GST), ubiquitin, ubiquitin-like modifier (SUMO), intrinsically disordered polypeptides, and the N-terminal domain of spider silk protein [145,146,147,148,149,150,151]. In this context, tags such as MBP are referred to as molecular chaperones [152]. Similarly, native molecular chaperones themselves can serve as effective solubility-enhancing tags in the fusion context [153,154,155,156]. Importantly, the robust effectiveness of the fusion approach strongly suggests that soluble-enhancing tags, including molecular chaperones, possess generic intrinsic chaperone-like activities for their tethered proteins. Despite its high efficacy and widespread popularity as a tool, it is surprising that this technique has long been regarded merely as an artificial technology, with little consideration given to its physiological relevance and significance. Artificial fusion proteins are essentially types of multidomain proteins where the N-terminal domains function as solubility-enhancing tags for downstream domains. This phenomenon is proposed to also occur in native multidomain proteins, as evidenced by observations that domains of native multidomain proteins can act as powerful solubility-enhancing tags for various aggregation-prone proteins in vivo [157]. The intrinsic chaperone-like activities of folded domains have been proposed to be attributed to the intermolecular repulsive forces generated by the surface charges and the excluded volume of the folded domains [157]. Multidomain proteins have been believed to be prone to aggregation due to their slow folding rates and tendency to misfold [10,63], consistent with the conventional paradigm. However, the intrinsic chaperone-like activities of individual domains, which can operate even without native interdomain interactions, suggest that multidomain proteins may be less prone to aggregation than previously believed [157]. Folded domains have been demonstrated to enable the independent folding of their neighboring domains through global phi value analysis [158]. Similar to ribosome tethering, these inconsistencies between protein folding and aggregation can also be explained by the independent effect of folded domains on both processes.

### 6.3. Artificial Chaperone System: Conversion of a Soluble Protein into a Potent Chaperone

The intrinsic chaperone-like activities of cellular macromolecules, as evidenced in the cis-tethering context, suggest that any soluble protein could potentially act as a potent chaperone if it can reversibly associate in trans with aggregation-prone proteins. As a proof of concept, a protein engineered with a protease mutant that recognizes a short seven-residue tag in trans has been transformed into a potent chaperone for various proteins bearing this tag, regardless of the tag’s position [159]. This artificial chaperone system exemplifies the intrinsic chaperone-like activity of a soluble protein functioning in trans. The chaperone-like activity of this protein is comparable to that of traditional molecular chaperones such as GroEL/ES, the DnaK system, and trigger factor in vivo. Notably, however, in the absence of aggregation, this protein exhibits no chaperone activity in vitro refolding experiments, as shown in Figure 7. These results indicate that the artificial chaperone increases the productive protein folding yield primarily by preventing aggregation without directly affecting the protein folding process itself. This also highlights the inconsistencies between protein folding and aggregation. Considering the ubiquitous physical linkage of proteins to various cellular macromolecules throughout their lifespan, these intrinsic chaperone-like activities significantly expand our understanding of how the cellular proteome maintains solubility against aggregation in the cellular milieu.

## 7. Aggregation Inhibition by Structure-Destabilizing Intermolecular Repulsive Forces

The new framework indicates that intermolecular repulsive forces play a crucial role in aggregation inhibition, independent of the intrinsic properties of proteins, such as protein folding, structure, and conformational change. This mode of aggregation inhibition is often overlooked or ignored in modern protein science. In this section, we explore the significant role of structure-destabilizing intermolecular repulsive forces, which may underpin the intrinsic aggregation inhibition capabilities of cellular macromolecules, including traditional molecular chaperones.

### 7.1. Pitfalls of Modern Protein Science

This section explains why modern protein science inherently offers a skewed perspective in addressing protein solubility maintenance against aggregation, particularly in the cellular milieu. The structure–function paradigm and structure-based drug design fundamentally depend on a detailed understanding of protein structures to elucidate functions and develop effective therapeutic agents. Thus, a defining trend in modern protein science focuses on ‘structural formation or molecular compaction’, including protein folding, native assembly, binding with interacting partners, and aggregation formation. For instance, the classic protein folding problem, structure predictions using AlphaFold and RoseTTAFold, and amyloid formation are all centered around the concept of structural formation [160,161,162,163]. Traditionally, structural formation has been predominantly understood and described through structure-stabilizing attractive forces. For instance, the energy landscape theory of protein folding, aggregation, and binding assumes that stabilizing attractive interactions in final structures dictate these landscapes [62,123]. Hydrophobic interactions as representative attractive interactions are presumed to govern substrate recognition and stabilization against aggregation in the mechanism of molecular chaperone action. Similarly, studies of aggregation have focused on ‘aggregation formation’ to find avenues to inhibit aggregation [163,164,165].

Solubility maintenance of polypeptides against aggregation (or aggregation inhibition) can be conceptualized as ‘anti-structural formation’, which serves as the opposite or reverse reaction to the structural formation, as depicted in Figure 8. Maintaining protein solubility against aggregation is achieved when aggregation-destabilizing forces and factors, such as intermolecular repulsive forces, desolvation penalties, and diverse entropic penalties, energetically outweigh aggregation-stabilizing forces and factors. Consistently, the solubility of non-protein molecules, including chemicals, other cellular macromolecules, and colloids, is predominantly explained by intermolecular repulsive forces, including electrostatic and steric repulsions [166,167]. Such observations underscore the stark contrast (or seemingly opposite approaches) approaches to managing aggregation between protein science and material science. Structure-stabilizing and structure-destabilizing forces are not mutually exclusive; they coexist. Real protein conformational landscapes are shaped by a balance between these forces from the intrinsic and extrinsic factors [32]. The structural formation, including aggregation formation, is driven by a protein’s inherent properties. However, anti-structural formation, such as aggregation inhibition and disaggregation, can be driven by a variety of extrinsic factors—such as cellular macromolecules—independently of the protein’s intrinsic properties, including protein folding, structure, and conformational change, as evidenced by the GroEL/ES chamber.

### 7.2. Structure-Destabilizing Intermolecular Repulsive Forces in Protein Folding

Structure-destabilizing forces are challenging to detect and quantify in final structures, yet they significantly influence structure formation landscapes [31,168]. The typical stability of proteins ranges from −5 to −15 kcal/mol [169]. Traditionally, the study of protein-folding landscapes has focused on stabilizing attractive interactions, such as hydrophobic interactions [62,170]. An unpaired hydrogen donor or acceptor in the protein core imposes an energetic penalty of approximately 5 kcal/mol, potentially leading to the global unfolding of proteins [171,172]. Similarly, a single hydrogen bond within the hydrophobic core of superoxide dismutase 1 (SOD1), implicated in ALS, is critical for affecting the global dynamics essential to the enzyme’s functionality [173]. Furthermore, a salt bridge in hydrophobic environments can induce an energy penalty of up to 19 kcal/mol [174,175], explaining why charged residues are typically located on protein surfaces. Steric clashes also play a crucial role in determining spatial constraints within protein structures, as evidenced by the Ramachandran plot, where the angles ϕ (phi) and ψ (psi) are restricted to avoid these clashes [176]. Such steric clashes result from the excluded volume repulsions of molecules. These direct structure-destabilizing intermolecular repulsive forces differ from conformational entropy, which quantifies the distribution of all possible conformational states of proteins. Protein folding occurs in the constraints imposed by these intermolecular repulsive forces [168,172]. These constraints may explain why a single point mutation can profoundly affect protein folding rate and stability, even though the final structures appear unchanged. George Rose’s remark, “Seeing is deceiving,” indicates that the visible final structures may not fully reflect the important role of structure-destabilizing repulsive forces, which are crucial for the actual protein folding landscapes [168]. His remark is also relevant to understanding protein solubility against aggregation (or anti-structural formation).

### 7.3. Role of Intermolecular Repulsive Forces in Aggregation Inhibition

From a simple physicochemical view, cellular macromolecules, with their numerous charges and large excluded volume, can be seen as scaled-up versions of small charged residues in proteins [31]. Empirically, it is widely accepted that individual charged residues of cellular macromolecules, including proteins, are important for maintaining their solubility against aggregation. Notably, however, little attention is given to the possibility that the exposed charges on cellular macromolecules might inhibit the aggregation of their physically connected polypeptides. Once physically connected, the complexes formed by cellular macromolecules and polypeptides can be considered as single entities, where the extrinsic charges of cellular macromolecules can act in a manner similar to intrinsic charges, thus influencing the aggregation behaviors of the connected polypeptides [31]. Consistently, the charges and hydrophilicity of solubility-enhancing tags play a crucial role in increasing the solubility of their connected aggregation-prone proteins [146,157,177,178,179,180,181]. Protein aggregation involves the assembly of multiple molecules and exhibits a degree of specificity [182,183]. Bulky cellular macromolecules with large excluded volumes likely inhibit the self-aggregation of their physically connected polypeptides. Importantly, it is a fact, not merely an assumption, that the excluded volumes of cellular macromolecules cannot be violated during the aggregation of their connected polypeptides. As illustrated in Figure 9, the cellular macromolecules have been proposed to generally exhibit the intrinsic chaperone activity to prevent the aggregation of their physically connected polypeptides, attributed to the intermolecular repulsive forces generated by their numerous surface charges and huge excluded volume [3,43,44,112,129,157,159,184,185]. This action mechanism is similar to those seen with charges as aggregation gatekeepers, the GroEL/ES chamber as the Anfinsen cage, and colloidal stability. Upon forming physical connections, the aggregation inhibition by cellular macromolecules can occur independently of the types of physical connections (whether in trans or in cis and whether hydrophobic or non-hydrophobic) and conformational changes. Of note, the repulsive forces exerted by cellular macromolecules are long-range due to their large size and multimolecular assembly, potentially affecting entire regions of the linked polypeptides [31]. This allows for the protection of aggregation-prone regions without the need for direct masking or reliance on attractive interactions. This long-range effect on aggregation inhibition is referred to as the allosteric effect of cellular macromolecules [159]. This stabilizing mechanism likely underlies the generic intrinsic chaperone-like activities in macromolecular tethering, fusion approach, and the artificial chaperone system. According to this model, gigantic and polyionic structures, such as ribosomes and lipid bilayers, could ideally serve as chaperone-like molecules in cis [43,45,129]. Similarly, intrinsically disordered, highly charged polypeptides function as effective solubilizers in the fusion context by creating extensive hydrophilic surface areas and large excluded volumes, termed entropic bristles [150]. Moreover, N-terminal protein tails act as protective entropic bristles that mitigate aggregation, as observed in the study of SUMO proteins [186].

### 7.4. The Role of Intermolecular Repulsive Forces in the Substrate Stabilization by Molecular Chaperones

Aggregation inhibition driven by intermolecular repulsive forces can also apply to substrate stabilization by traditional molecular chaperones. DnaK, the *E. coli* Hsp70 homolog, binds hydrophobic-rich peptides (e.g., NRLLLTG) [55], but it masks only a small part of the protein’s hydrophobic surfaces. In the fusion context, mutation or deletion in DnaK’s substrate binding residue or domain does not compromise its ability to prevent the aggregation of fused proteins, suggesting that the primary stabilization mechanism may be through the repulsive forces from DnaK’s surface charges and large excluded volume rather than limited hydrophobic interactions [184]. The anti-aggregation ability of Hsp90 relies on two charged patches. Removing these patches significantly diminishes its aggregation inhibition capabilities, although it does not affect substrate binding. However, reintroducing an artificial acid-rich sequence restores this function, emphasizing that the net charge of Hsp90 is crucial for its aggregation inhibition activity [187]. These observations show that the extrinsic surface charges of cellular macromolecules can inhibit aggregation of connected polypeptides across significant distances, consistent with the aforementioned allosteric effect of cellular macromolecules. Similarly, the extensively charged poly-Asp/Glu region of the DAXX chaperone is crucial for its diverse activities, including aggregation inhibition and disaggregation [188]. The physical basis for the direct steric masking by GroEL/ES and TriC/CCT, which inhibits aggregation, is attributed to the excluded volume repulsions exerted by these chaperonins. Similarly, the diverse functions of Hsp70, such as translocation and disaggregation, can be universally explained by entropic pulling forces rooted in its excluded volume repulsions [189]. Furthermore, the direct steric repulsions between DnaK molecules, when bound to a misfolded client, are suggested to play a role in rescuing misfolding [65,190].

The stabilization of substrates by intermolecular repulsive forces suggests that molecular chaperones can interact with their clients through various non-hydrophobic interactions, thereby broadening their functional capabilities. Consistently, molecular chaperones, including GroEL, TriC/CCT, trigger factor, Spy, and DAXX, can predominantly recognize their clients through electrostatic interactions [188,191,192,193]. This mode of binding via electrostatic interactions offers a significant advantage, as it can facilitate the proper folding of proteins in the binding context [194]. In this regard, macromolecular tethering can allow cotranslational (cotranslocational) protein folding and domain folding in multidomain proteins to freely occur while independently and simultaneously inhibiting aggregation [43,45,129]. This scenario is similar to protein folding in the GroEL/ES chamber, while aggregation is completely blocked. Calnexin and calreticulin recognize the glycan moieties of their clients [195]. The substrate stabilization by these recognition types, akin to that seen with the artificial chaperone system [159], appears difficult to explain solely through direct attractive interactions. In contrast, the substrate stabilization facilitated by intermolecular repulsive forces can occur regardless of the types of physical linkage. This scenario suggests that calnexin and calreticulin are expected to allow protein folding to proceed in the binding context while preventing aggregation. A variety of non-hydrophobic attractive interactions, including hydrogen bonding, van der Waals interactions, and electrostatic interactions, are significantly involved in protein aggregation formation. Aggregation driven by non-hydrophobic interactions is difficult to explain solely through hydrophobic interactions between molecular chaperones and their clients.

The new framework suggests a unifying mechanism—intermolecular repulsive force-driven aggregation inhibition—that underpins the diverse sources of aggregation inhibition. These sources include ATP-dependent and -independent molecular chaperones, macromolecular tethering, fusion approaches, the artificial chaperone system, calnexin/calreticulin, and charges as aggregation gatekeepers, along with colloidal stability. Taken together, it becomes apparent that the prevailing interpretation of molecular chaperone functions, traditionally based solely on hydrophobic interaction-mediated substrate recognition and stabilization, may be narrow. In reality, quantifying the involved stabilizing forces against aggregation remains almost completely uncharted territory in the field of protein science. Even the stabilizing forces in protein folding, which remain controversial, have yet to be fully elucidated [170,171,172]. Therefore, maintaining an open mind is necessary as it enables a broader exploration of how the cellular proteome maintains solubility against aggregation in the complex cellular milieu.

## 8. Metastability Revisited Through the New Framework

Metastability describes a state in which molecular systems persist due to high kinetic barriers that inhibit their transition to more thermodynamically stable states [196]. Traditionally, folded and functional protein structures are considered metastable in relation to aggregates [6,21,22,23]. Aggregation processes are typically nonequilibrium, leading to poorly characterized aggregation thermodynamics experimentally; therefore, they are generally interpreted kinetically. Notably, however, the presence of aggregates in nonequilibrium conditions does not necessarily imply that they are thermodynamically more stable than soluble monomers. As detailed in Section 4, the transition from folded structures to aggregates involves both intramolecular protein folding thermodynamics and intermolecular aggregation thermodynamics, which are independent yet interconnected via shared aggregation monomers (Figure 2 and Figure 4). The shared aggregation monomers, such as misfolded or unfolded monomers, serve as the main reactants in equilibrium with aggregates within aggregation thermodynamics; their relationship determines the aggregation free energy landscape. Folded structures and shared aggregation monomers can exhibit different intramolecular thermodynamic stabilities and markedly different aggregation behaviors, requiring a clear differentiation between these forms. The concentration of denatured forms is used to calculate aggregation thermodynamics relative to amyloid fibrils [78,197]. The aggregation thermodynamics of unfoldable polypeptides alone, such as IDPs and IDRs, is unrelated to folded structures. Therefore, understanding metastability needs to be understood in the context of shared aggregation monomers.

### 8.1. Strategies for Decreasing the Effective Concentration of Shared Aggregation Monomers

In contrast to intramolecular protein folding, where thermodynamics is determined by the N/D ratio that remains unaffected by actual protein concentration, aggregation (or solubility) thermodynamics is significantly influenced by this concentration. Supersaturation occurs when the protein concentration exceeds its equilibrium solubility limit in a solvent, resulting in a metastable state where the system is prone to spontaneous aggregation or crystallization as it seeks a free energy minimum [198]. Supersaturation is quantified by the supersaturation ratio: S = *C*/*C*_sat_, where *C* is the initial protein concentration, and *C*_sat_ is its solubility limit [78,199]. If S > 1, proteins are in a supersaturated, metastable state, predisposed to aggregation. Protein concentrations in vivo can exist near their solubility limits at the proteome level [200,201]. Notably, according to the new framework, the *C* and *C*_sat_ in the supersaturation formula *S* specifically refer to the concentration of shared aggregation monomers, not to the total soluble monomer protein concentration that includes folded structures. More precisely, these values represent the thermodynamic activity or the effective thermodynamic concentration of shared aggregation monomers.

Strategies aimed at decreasing the effective concentration of shared aggregation monomers could lead to relative thermodynamic stabilization of soluble monomers against aggregation, as illustrated in Figure 10A. Firstly, significant amounts of molecular chaperones, including protein quality control systems, are present at the proteome level in vivo [202]. Their complexation with shared aggregation monomers shifts the equilibrium, favoring a reduction in the concentration of free shared aggregation monomers. Secondly, folded structures and their native complexes with interactors—such as cellular macromolecules and metals—can dramatically decrease the concentration of shared aggregation monomers. Importantly, proteins undergo extensive native complexation in vivo and do not exist in isolation. For instance, although ribosomal S6 proteins alone can fold efficiently with marginal stability [38], they are primarily found within ribosome complexes. This crucial aspect of native complexation is often overlooked in discussions of protein metastability. Thirdly, macromolecular tethering is ubiquitous in vivo at the proteome level, including ribosomes, multidomain proteins, lipid bilayers, and post-translational modifications such as ubiquitination, sumoylation, and glycosylation [32]. If macromolecular tethering inhibits the self-aggregation of tethered polypeptides, it can potentially reduce the effective concentration of shared aggregation monomers. The covalent or covalent-like linkage in macromolecular tethering can be considered an irreversible, tight binding in trans. Overall, the effective thermodynamic concentration of free, shared aggregation monomers in vivo could be significantly lower than previously anticipated due to their physical linkage with cellular macromolecules or ligands. Consequently, these free soluble monomers might be thermodynamically more stable than amyloid fibrils in vivo. By recognizing shared aggregation monomers as the principal reactants in aggregation, we can see that the conventional methods of aggregation inhibition using molecular chaperones and macromolecular tethering, typically viewed solely in terms of kinetic barriers, might also serve as thermodynamic stabilizers against aggregation [31]. Consistently, kinetic factors could be thermodynamic factors in dynamic equilibrium conditions.

### 8.2. Thermodynamic Destabilization of Aggregates by Physical Linkage of Cellular Macromolecules

In thermodynamics, changes in Gibbs free energy reflect the relative differences in energy levels between reactants and products. This complexity complicates precise assessments of how specific factors influence the energy levels of aggregation monomers or their aggregates. Keeping the ground states of aggregation monomers constant can help evaluate these impacts more clearly.

Using a macromolecular tethering model, it has been suggested that such tethering can thermodynamically destabilize aggregates, such as amyloids, by elevating their ground state energy [31], as depicted in Figure 10B. This model employs a two-state folding approach in which macromolecular tethering does not alter the protein folding itself, thereby leaving the energy levels of the folded and unfolded states (assumed as soluble aggregation monomers here) unchanged. However, macromolecular tethering can destabilize the self-aggregation of tethered proteins through their intermolecular repulsions caused by their surface charges and excluded volumes, thereby increasing the ground state energy levels of the aggregates. Similarly, molecular chaperones are proposed to thermodynamically destabilize amyloids by forming coaggregates with amyloid peptides, thereby enhancing the solubility of monomers [203]. This coaggregation is similar to the aggregation of polypeptides tethered to cellular macromolecules. Remarkably, proteins with more than 150 amino acids can be inherently more thermodynamically stable due to their increased size, suggesting that larger proteins are better equipped to overcome the metastability challenges associated with smaller peptides [6]. Consistently, the aggregation of small peptide fragments is associated with proteinopathies [6]. A common feature among macromolecular tethering, coaggregation with molecular chaperones, and increased protein size is the physical linkage of thermodynamically aggregation-prone polypeptides alone with other large molecular regions. The relative monomer stabilization against aggregation formation (or anti-structural formation) results from the intrinsic properties of these regions themselves, akin to charges as aggregation gatekeepers and the GroEL/ES chamber. These phenomena are challenging to explain with the conventional paradigm, the fundamentals of modern protein science, and the traditional concept of metastability. In contrast, the new framework provides a coherent explanation by applying its principles to metastability.

## 9. Applications of the New Framework to Therapeutic Interventions for Proteinopathies

The conventional paradigm in protein science, which does not strictly distinguish between protein folding (or misfolding) and aggregation, has long shaped our understanding and treatment strategies for proteinopathies. Recognizing the independence between misfolding and aggregation, alongside the intrinsic chaperone-like activities of cellular macromolecules, the intermolecular repulsive force-driven aggregation inhibition, and their long-range allosteric effect, provide new insights for developing therapeutic interventions.

### 9.1. Targeting Cellular Macromolecules as Drug Candidates in Proteinopathies

Cellular macromolecules, whether directly or indirectly associated with proteins or peptides involved in disorders, can prevent aggregation through their intrinsic chaperone-like activities. All these cellular macromolecules could represent potential targets for therapeutic interventions, as illustrated in Figure 11A. Given the ongoing challenges and the persistent lack of effective treatments despite decades of research targeting these cellular macromolecules offers significant potential for breakthroughs. Consistently, the flanking regions of pathogenic polypeptides have a significant impact on aggregation behaviors [204,205,206,207,208,209]. Additionally, small heat-shock proteins binding to flanking domains can inhibit the aggregation of polyglutamine repeat expansions, which are associated with disorders such as Huntington’s disease and spinocerebellar ataxias, even though these proteins do not directly interact with the aggregation-prone regions [210]. Disease-associated polypeptides are connected with a variety of cellular macromolecules, forming complex interactomes [211,212,213,214,215]. For instance, amyloid beta peptides (Aβ) in amyloid precursor proteins (APP) are physically tethered to other regions of APP and to cellular membranes [216]. APP forms a physical interactome to fulfill its functional roles [217]. Aβ, once cleaved from APP, interacts with a variety of proteins, including molecular chaperones [218]. Small molecules enhancing these interactions or stabilizing the macromolecules are proposed to serve as potential drug candidates [159]. Bifunctional small molecules that bind both chaperones and amyloid fibrils harness the steric bulkiness of chaperones to effectively inhibit Aβ aggregation [219]. In this approach, chaperones can be replaced with other cellular macromolecules since steric bulkiness is a common feature of cellular macromolecules with large excluded volumes. Mutations in charged residues of proteins are closely associated with proteinopathies [220,221,222,223]. Modulating amyloid formation by altering the net charges of proteins through chemical intervention is a promising strategy [224]. This approach aligns with the intrinsic chaperone activities of cellular macromolecules, which harness their surface charges to inhibit aggregation.

### 9.2. Cellular Macromolecules as Native Interactors: Promising Drug Targets in Proteinopathies

Cellular macromolecules that function as native interactors could be promising drug targets. Most cellular proteomes are involved in native interactions or complexes with proteins, nucleic acids, and metals [116,117,118,119,120,124,225]. Cellular macromolecules as native interactors can significantly influence the folding and stability of associated pathogenic polypeptides, as depicted in Figure 11B. Moreover, they can also independently exhibit intrinsic chaperone-like activities. The proper stoichiometry of these native complexes is crucial, and disruptions to this balance can occur with aging, potentially leading to disease progression [226,227]. Consistently, specific polypeptides are associated with particular proteinopathies [16]. For instance, AD is characterized by Aβ and tau proteins, prion diseases by prion protein, and Huntington’s disease by huntingtin protein. This specificity could be attributed to a lack or impairment of physical interactions with specific native interactors. The binding-coupled protein folding of IDPs is ubiquitous, and their folding is strictly dependent on their native interactors [228]. Most pathogenic polypeptides are either IDPs or IDRs [229]. IDPs and IDRs pose a challenge in drug development due to their lack of a druggable stable pharmacophore, making it difficult to target them effectively with traditional small molecule drugs [230]. Therefore, targeting cellular macromolecules as native interactors represents a promising strategy for modulating protein folding, stability, the stoichiometry of complexation, and aggregation. So far, however, according to the amyloid cascade hypothesis [231], drug development has primarily focused on preventing the aggregation of pathogenic polypeptides themselves. Furthermore, the molecular chaperone concept traditionally overlooks the role of native interactors, as these ligands are not classified as molecular chaperones. In contrast, small chemical compounds known as pharmacological chaperones bind to native structures and stabilize them, functioning similarly to native interactors or ligands [232,233]. Drugs that stabilize native complexation can prevent the formation of aggregation monomers at an upstream level. These considerations reinforce that cellular macromolecules as native interactors are promising drug targets.

## 10. Conclusions and Future Perspectives

We introduce a new framework that redefines the relationship between intramolecular protein folding and intermolecular aggregation as independent yet interconnected via shared aggregation monomers. The new framework extends the conventional paradigm that views aggregation primarily as a direct outcome or continuum of protein folding and misfolding. It establishes that intrinsic and extrinsic factors independently affect both processes. Furthermore, it highlights that intermolecular aggregation inhibition by intermolecular repulsive forces can operate independently of protein folding, structure, and conformational change. The new framework revisits and provides new insights into the fundamental issues at the interface between protein folding and aggregation.

The new framework provides a more comprehensive understanding of the relationship between protein folding and aggregation within the cellular milieu, where polypeptides are physically connected to a variety of cellular macromolecules throughout their lifespan. Moreover, it can be applied to the relationships between diverse protein conformational landscapes, with each independent landscape interconnected through shared structures. Particularly, the framework emphasizes the necessity of considering and exploring structure-destabilizing intermolecular repulsive forces, as real protein conformational landscapes within the cellular milieu are shaped by a delicate balance between structure-stabilizing and destabilizing forces arising from the intrinsic properties of proteins and extrinsic cellular factors. In this context, the role of the cellular milieu, particularly in protein aggregation, needs to be redefined. Finally, the new framework offers valuable insights that can enhance our understanding of complex phenomena, including combined protein folding and aggregation, proteostasis, and proteinopathies.

## Figures and Tables

**Figure 1 ijms-26-00053-f001:**
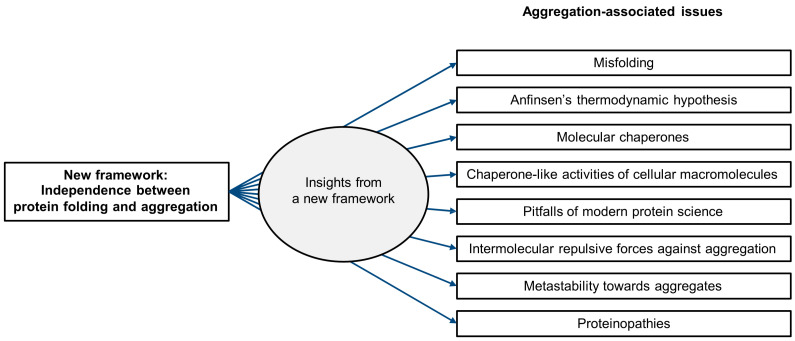
A new framework for the relationship between protein folding and aggregation: Its implications for aggregation-associated issues. This framework redefines the relationship between protein folding and aggregation, distinguishing them as independent yet interconnected processes, in contrast to the conventional paradigm that treats aggregation primarily as a consequence of protein folding and misfolding. Through this framework, the fundamental issues at the interface between protein folding and aggregation are reevaluated.

**Figure 2 ijms-26-00053-f002:**
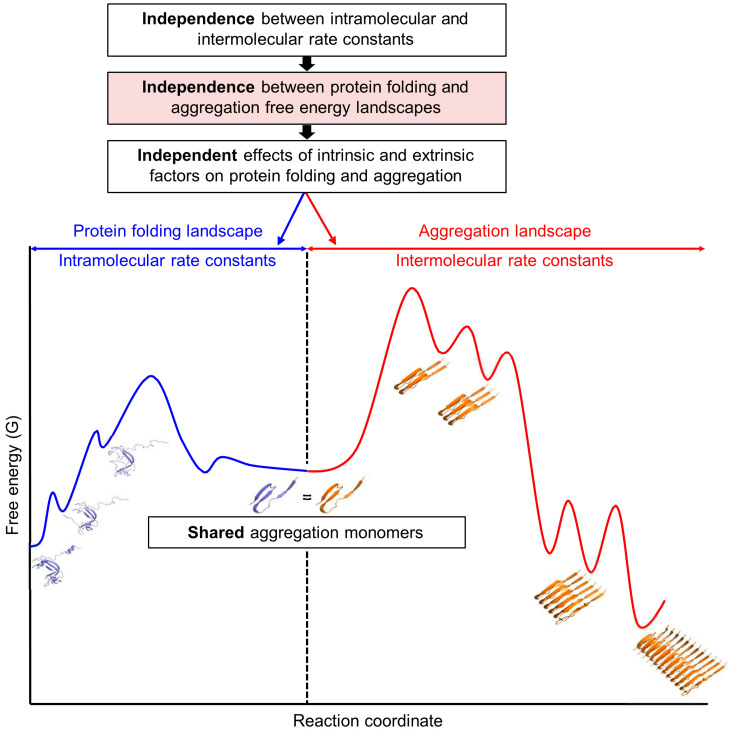
Independence between protein folding and aggregation. Intramolecular and intermolecular rate constants are mutually independent, shaping independent free energy landscapes for protein folding and aggregation, respectively. Consequently, intrinsic and extrinsic factors influence these landscapes independently. Despite their independence, both landscapes are interconnected through shared aggregation monomers, which can exist in various structural states, including folded, unfolded, misfolded, or intermediate forms. Of note, the free energy landscapes depicted in Figures 2–11 are oversimplified; the relative ground state energy levels and the heights of kinetic barriers are conceptual representations, not quantitative measurements.

**Figure 3 ijms-26-00053-f003:**
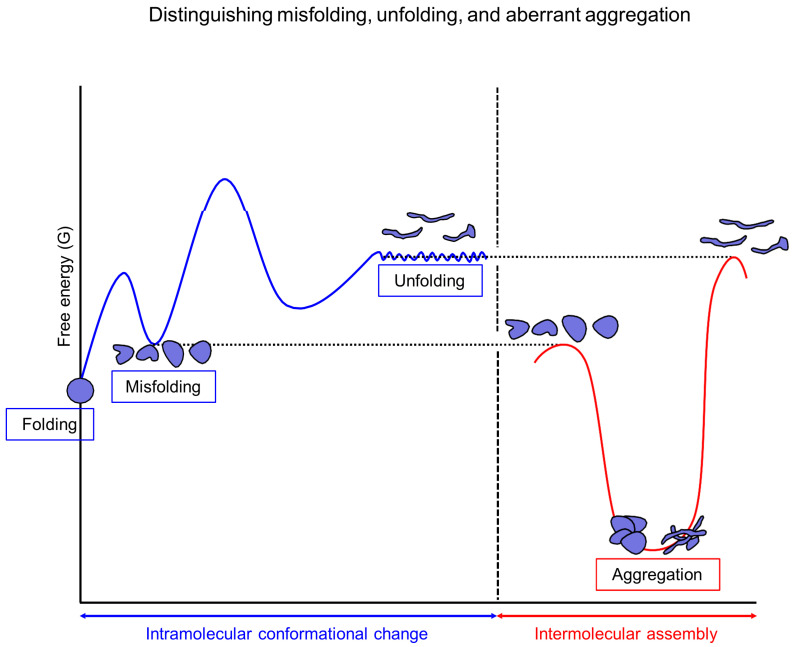
Distinguishing misfolding, unfolding, and aggregation. Misfolding occurs when proteins adopt non-native structures that become kinetically trapped near their native states. Unfolding, in contrast, involves a transition to higher-energy unfolded structures that are not kinetically trapped at the intramolecular level, with conversions between these ensembles occurring rapidly. Aberrant aggregation, distinct from intramolecular misfolding and unfolding, represents the intermolecular assembly of various non-native forms, not necessarily limited to misfolded structures.

**Figure 4 ijms-26-00053-f004:**
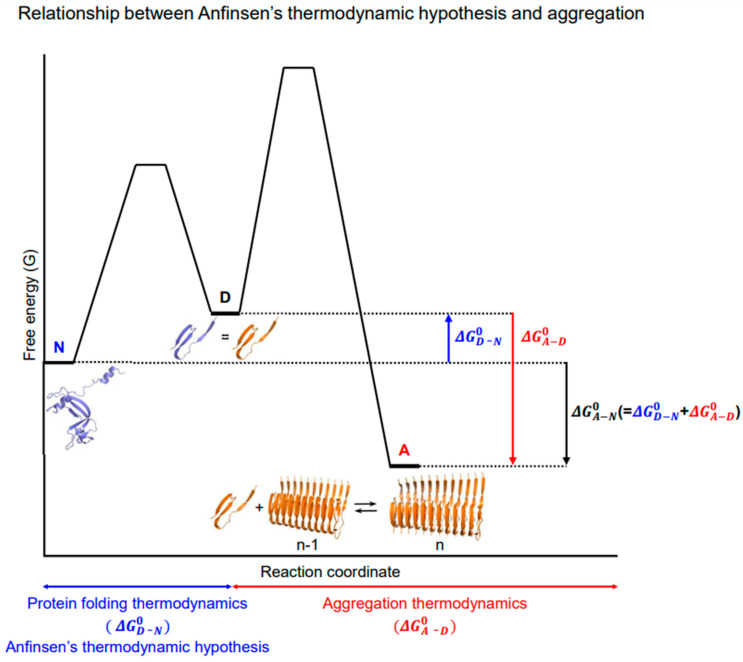
Relationship between Anfinsen’s thermodynamic hypothesis and aggregation. Anfinsen’s thermodynamic hypothesis only applies to the intramolecular conformational changes and protein folding thermodynamics (Δ$G_{D-N}^{0}$) between folded (N) and denatured (D) states. The intermolecular aggregation thermodynamics (Δ$G_{A-D}^{0}$) specifically relates to the transition from shared aggregation monomers (D), serving as the main reactants, to their resultant aggregates (A). The standard Gibbs energies (Δ$G_{D-N}^{0}$) for protein unfolding and (Δ$G_{A-D}^{0}$) for aggregation, determined by their respective rate (or equilibrium) constants under standard conditions, are independent. Δ$G_{A-N}^{0}$ between folded to aggregated states can be indirectly obtained by combining Δ$G_{D-N}^{0}$ and Δ$G_{A-D}^{0}$.

**Figure 5 ijms-26-00053-f005:**
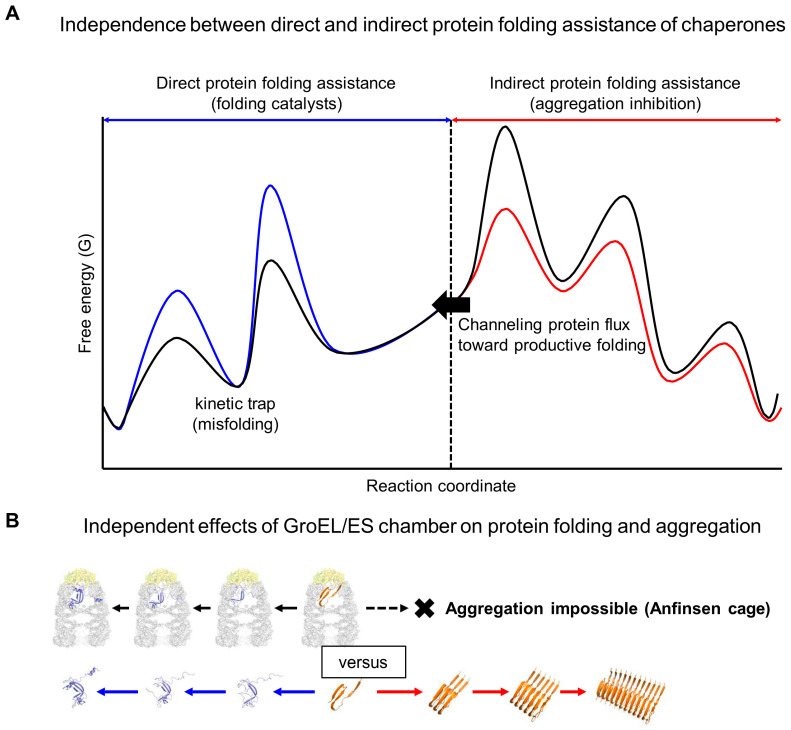
(**A**) Independence between direct and indirect protein folding assistance by molecular chaperones. Molecular chaperones function as folding catalysts, directly assisting protein folding. They also provide indirect assistance by preventing off-pathway aggregation, thus channeling protein flux towards productive folding pathways, as indicated by the black arrow. Both direct and indirect assistance are independent actions of molecular chaperones. (**B**) Independent effects of GroEL/ES chamber on protein folding and aggregation. The GroEL/ES chamber, known as the Anfinsen cage, completely prevents aggregation while independently affecting protein folding in it. Detailed explanation is available in the text.

**Figure 6 ijms-26-00053-f006:**
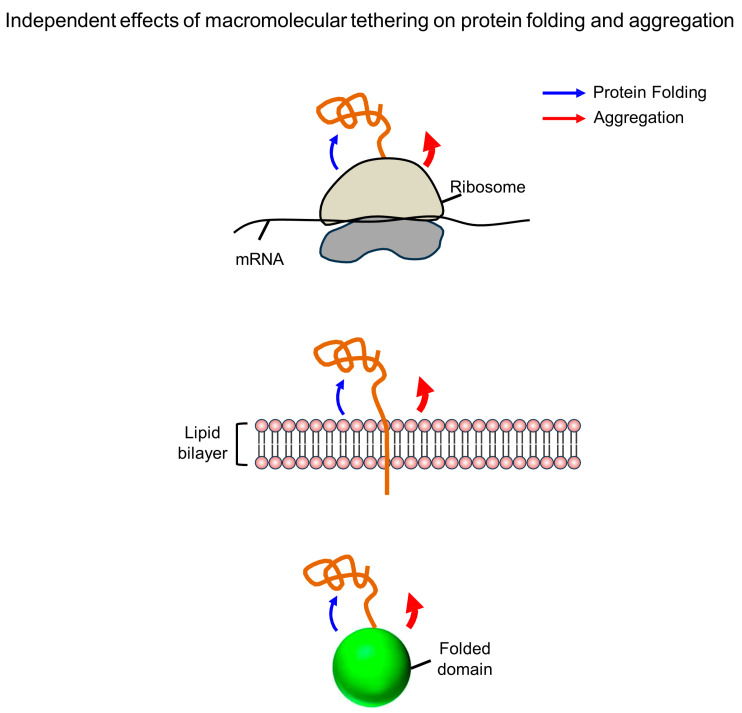
Independent effects of macromolecular tethering on protein folding and aggregation. Macromolecular tethering is a fundamental aspect of de novo protein folding; newly synthesized polypeptides are tightly tethered to ribosomes (representing the whole proteome), lipid bilayers (30% of the proteome), or other domains in multidomain proteins (80% of the proteome). The new framework suggests that macromolecular tethering independently affects intramolecular protein folding (blue arrows) and intermolecular aggregation (red arrows).

**Figure 7 ijms-26-00053-f007:**
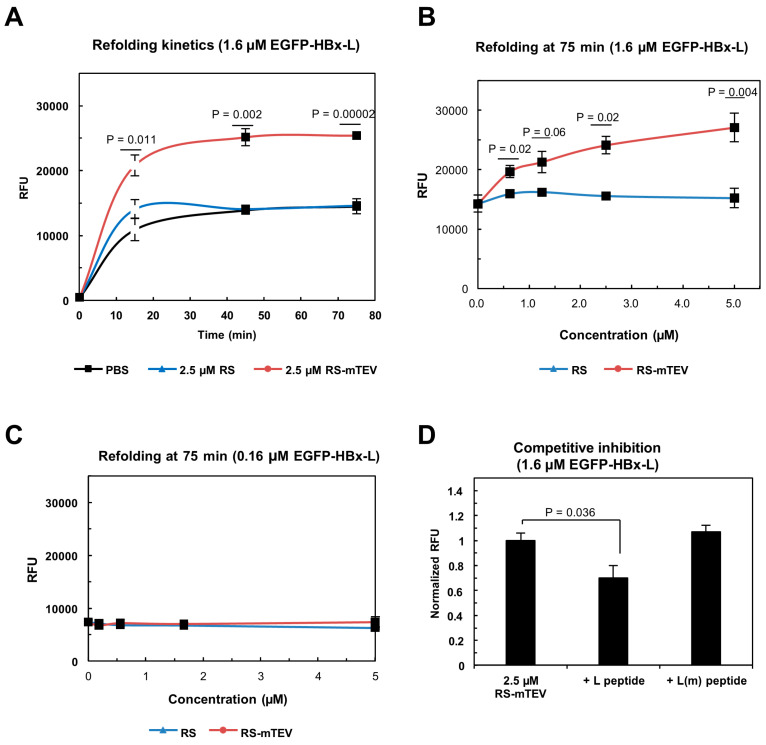
Independent effects of artificial chaperone on protein folding and aggregation. The artificial chaperone (RS-mTEV) enhances productive refolding yield in the presence of aggregation (panels **A**,**B**) but exhibits no chaperone activity in the absence of aggregation (panel **C**). Its chaperone activity is specifically inhibited by the substrate recognition peptide (L) but not by a non-competing peptide [L(m)] (panel **D**). RS-mTEV was engineered by linking a mutant TEV protease domain, capable of binding a specific 7-residue recognition sequence (L), to the C-terminus of *E. coli* LysRS. The client protein EGFP-HBx-L consists of enhanced green fluorescent protein (EGFP), hepatitis B virus x protein (HBx), and the recognition sequence (L). This figure is adapted from Ref. [159].

**Figure 8 ijms-26-00053-f008:**
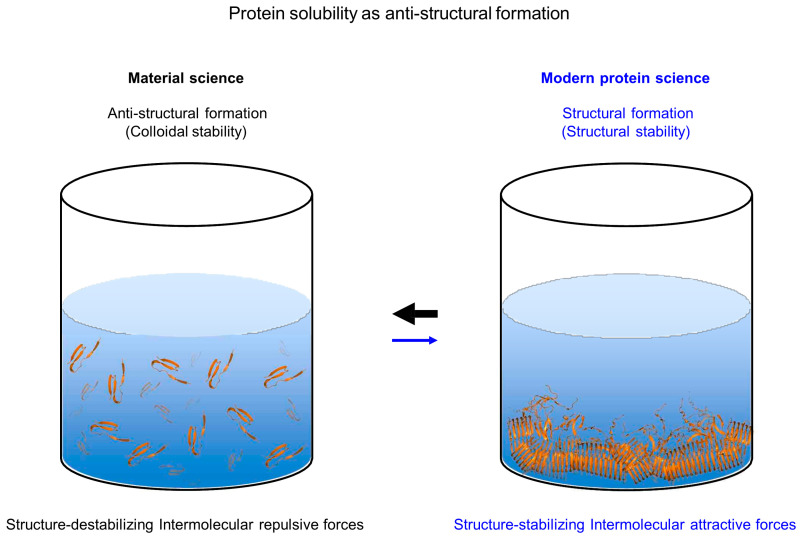
Understanding protein solubility maintenance against aggregation as anti-structural formation. This figure contrasts the traditional protein science approach, which focuses on structural formation through attractive interactions, with material science’s approach, which emphasizes anti-structural formation driven by intermolecular repulsive forces to maintain protein solubility. Protein solubility is maintained when the aggregation-destabilizing forces and factors energetically dominate over the aggregation-stabilizing attractive forces and factors.

**Figure 9 ijms-26-00053-f009:**
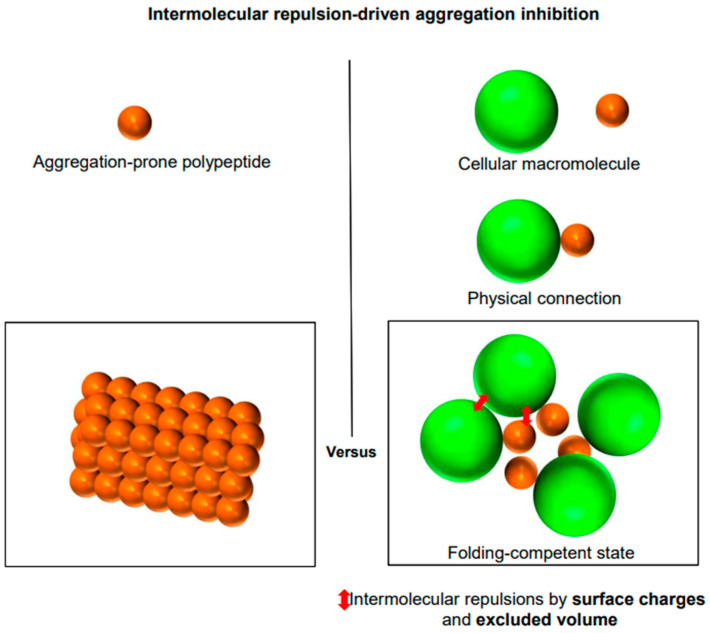
Intermolecular repulsive force-driven aggregation inhibition. Cellular macromolecules (green spheres) exhibit large excluded volumes and numerous surface charges, generating intermolecular repulsions (red arrows) against the aggregation of their physically connected polypeptides (orange speres). This figure is adapted from Ref. [3].

**Figure 10 ijms-26-00053-f010:**
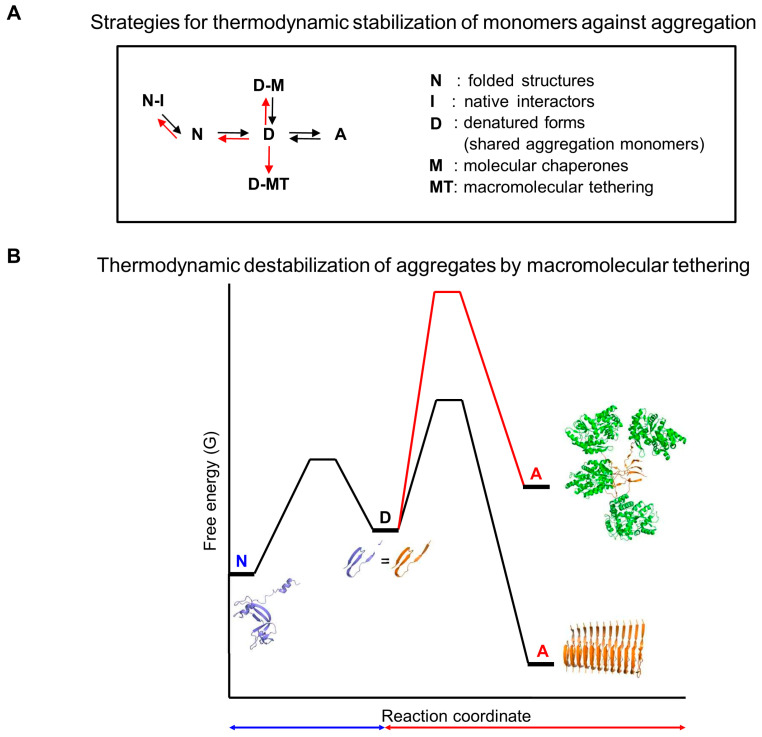
Thermodynamic stabilization of monomers against aggregation. (**A**) Routes for decreasing effective concentration of shared aggregation monomers. The effective concentration of aggregation monomers is reduced by the physical linkage with molecular chaperones (M), macromolecular tethering (MT), and native interactors (I). (**B**) Thermodynamic destabilization of aggregates by macromolecular tethering. The cellular macromolecules (green) destabilize the self-aggregates of tethered polypeptides, increasing the ground energy levels of aggregates without affecting the folding processes of tethered polypeptides.

**Figure 11 ijms-26-00053-f011:**
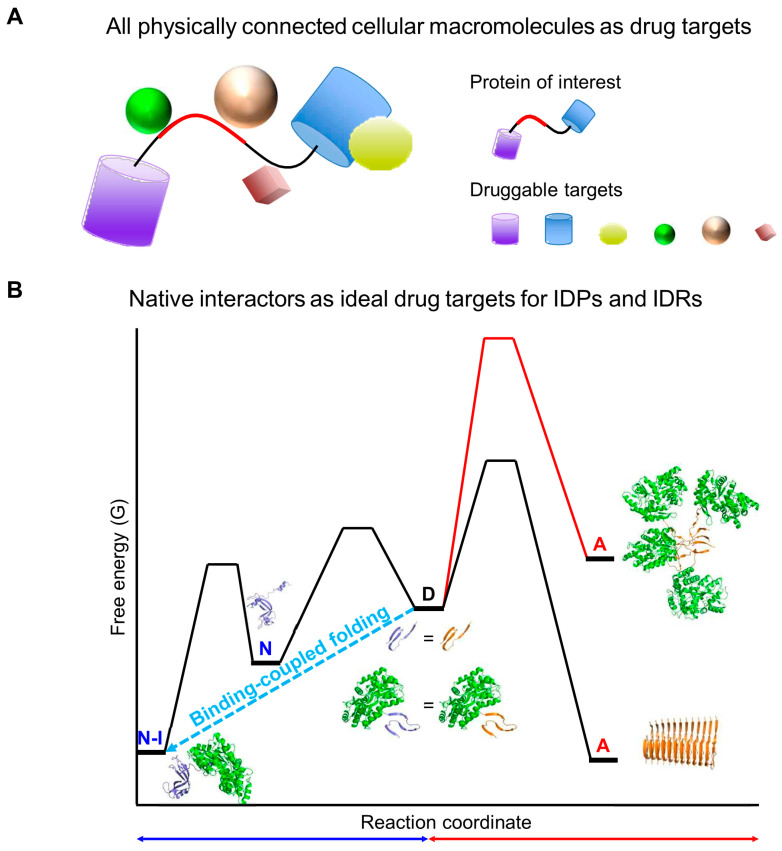
Therapeutic interventions for proteinopathies harnessing cellular macromolecules. (**A**) Cellular macromolecules as drug targets. All cellular macromolecules that are physically connected to pathogenic peptides (red) could be drug targets due to their intrinsic chaperone-like activities. (**B**) Native interactors as ideal drug targets. Native interactors play a crucial role in protein folding and stability of their cognate polypeptides while they exhibit intrinsic chaperone-like activities against aggregation. In particular, IDPs and IDRs undergo folding upon binding to their native interactors that independently exert chaperone-like activities against aggregation, suggesting that these native interactors could be ideal drug targets.

## Data Availability

Not applicable.

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
