# Peer review of "Beyond Misfolding: A New Paradigm for the Relationship Between Protein Folding and Aggregation"

_ijms, 2024, doi:10.3390/ijms26010053_

Round 1
Reviewer 1 Report
Comments and Suggestions for Authors
The paper by Choi et al. represents a comprehensive review of protein folding, misfolding, and unfolding, including the role of molecular chaperones and other cell components in these processes. The concept that unfolded states are common to the three phenomena is interesting and well justified in the paper, although I am not sure if the intrinsically disordered regions of proteins could not act as seeds of aggregation even if such folded proteins occur at higher concentrations. The authors should address this question.
The manuscript is generally well written but a couple of points need clarification.
1. In all diagrams the authors put the aggregate state (in the absence of destabilizing agents) lower in free energy than the folded state and the barrier to aggregation higher than that to the folded state. Some explanation as to why is the free energy of the aggregate lower and the free energy barrier higher is needed.
2. Page 4, Figure 1.: This diagram seems to be a collection of separate issues of different origin and meaning and it is not clear why they are put in the particular order. A better diagram or explanation is needed; a linked graph could probably do better.
3. Page 9, lines 377 amd 378: "The standard Gibbs energies for protein folding and aggregation are independent.": what does the "independence" of these two free energies mean?
4. Page 9, line 331: "D = Ctotal/(K +1)": K appears to be the equilbrium constant for the unfolded-folded state conversion. This explanation should be included.
5. Page 9, lines 332-334: "This equation shows that the intramolecular protein folding thermodynamics affect aggregation by determining the concentration of these aggregation monomers."
Probably: "...by determining the concentration of unfolded monomers."
6. Page 10, lines 339-340: "According to the independence between Anfinsen’s thermodynamic hypothesis and aggregation". This sentence is not clear: what does "independence" mean in this context?
7. Page 10, line 351: "rate constants, which operate independently" The "rate constants" cannot "operate". Please rephrase.
8. "The reaction Gibbs energy for aggregation reflects deviations from standard concentrations used in standard Gibbs energy calculations, thereby determining whether proteins are metastable or not."
Probably: "Gibbs free energy". Moreover, the sentence is not clear and needs rephrasing.
Author Response
We sincerely thank all the reviewers for their insightful and constructive comments, which have been crucial in enhancing both the quality and depth of our manuscript.
To facilitate the review process, in the revised manuscript, all modified or newly added texts are marked in red, and the flanking regions of deleted sections are highlighted in blue for enhanced clarity.
Please note that although page and line numbers of the revised manuscript might change upon uploading to the IJMS system, the color markings will ensure that revisions are easily identifiable.
Reviewer 1
Comments and Suggestions for Authors
The paper by Choi et al. represents a comprehensive review of protein folding, misfolding, and unfolding, including the role of molecular chaperones and other cell components in these processes. The concept that unfolded states are common to the three phenomena is interesting and well justified in the paper, although I am not sure if the intrinsically disordered regions of proteins could not act as seeds of aggregation even if such folded proteins occur at higher concentrations. The authors should address this question.
Response to Reviewer's Comment: Thank you for your comments. To clarify, our model indeed considers all intramolecular structures as shared aggregation monomers, including misfolded, unfolded, intermediate, and even folded structures, as described in our manuscript: “These monomers can exist in various states, including unfolded, misfolded, partially folded, or fully folded forms.” (page 6, line 161-162 in the Section 2). Accordingly, our concept of shared aggregation monomers encompasses folded proteins with intrinsically disordered regions (IDRs) that can act as seeds of aggregation.
Furthermore, unfolded structures are used as representative shared aggregation monomers to facilitate a direct comparative analysis of protein folding energetics relative to aggregation energetics throughout our manuscript. This approach is explicitly noted in the revised manuscript: “Throughout this paper, unfolded structures are used as representative shared aggregation monomers unless otherwise stated.” (page 6, line 162-164 in the Section 2).
The manuscript is generally well written but a couple of points need clarification.
1. In all diagrams the authors put the aggregate state (in the absence of destabilizing agents) lower in free energy than the folded state and the barrier to aggregation higher than that to the folded state. Some explanation as to why is the free energy of the aggregate lower and the free energy barrier higher is needed.
Response to Reviewer's Comment: Thank you for your question. Protein aggregates, such as amyloid fibrils, are generally believed to be thermodynamically more stable than folded states, since they are stable under high concentration of denaturants where folded structures are readily unfolded. However, the transition to these aggregates is inhibited by high kinetic barriers. For instance, the aggregation rate of amyloid formation can take several hours or days, which is several magnitudes higher than the folding (unfolding) rate that typically ranges from milliseconds to seconds. Consequently, we depict the aggregate state with lower free energy but higher energy barriers in our diagrams. However, it is important to emphasize that these diagrams are conceptual rather than quantitative, and the relative positions of ground states and the heights of barriers are subject to variation. This point has been newly clarified in the legend of Figure 2: “Of note, the free energy landscapes depicted from Figures 2 to Figure 11 are oversimplified; the relative ground state energy levels and the heights of kinetic barriers are conceptual representations, not quantitative measurements.” (page 45, line 1550-1553).
2. Page 4, Figure 1.: This diagram seems to be a collection of separate issues of different origin and meaning and it is not clear why they are put in the particular order. A better diagram or explanation is needed; a linked graph could probably do better.
Response to Reviewer's Comment: Thank you for your feedback regarding Figure 1. Based on reviewer’s recommendations, we have reorganized the elements within Figure 1 to more clearly illustrate the various aggregation-associated issues reassessed by our new framework. This revision enhances the diagram’s clarity.
3. Page 9, lines 377 amd 378: "The standard Gibbs energies for protein folding and aggregation are independent.": what does the "independence" of these two free energies mean?
Response to Reviewer's Comment: Thank you for your question. Throughout our manuscript, when we describe the standard Gibbs energies for protein folding and aggregation as "independent," it means that these values are calculated from independent sets of parameters that do not influence each other. In scientific terms, 'independence' implies that two or more variables or parameters can be altered or controlled independently, without any effect on one another. Consequently, it is impossible to predict the standard Gibbs energy for protein folding based on the standard Gibbs energy for aggregation, and vice versa.
More specifically, the equilibrium constants of intramolecular protein folding are independent of those of intermolecular aggregation equilibrium constants. Consequently, the "standard" Gibbs energies for protein folding and for aggregation, calculated from their respective equilibrium constants under standard conditions (where reactants and products are fixed at 1 mole each), are mutually independent.
4. Page 9, line 331: "D = Ctotal/(K +1)": K appears to be the equilibrium constant for the unfolded-folded state conversion. This explanation should be included.
Response to Reviewer's Comment: Thank you for your feedback regarding our description of the equilibrium constant. We have added a clarifying sentence in the manuscript to better define K as the equilibrium constant for the conversion between unfolded and folded states of the protein: “ D = Ctotal/(K +1), where Ctotal represents the given initial total protein concentration of soluble monomers and K is the equilibrium constant for protein folding (78).” (page11, Line 309-310 in the Section 4).
5. Page 9, lines 332-334: "This equation shows that the intramolecular protein folding thermodynamics affect aggregation by determining the concentration of these aggregation monomers." Probably: "...by determining the concentration of unfolded monomers."
Response to Reviewer's Comment: According to the reviewer’s comments, we have revised the sentence: “by determining the concentration of these unfolded monomers, which act as the main reactants for aggregation, in line with the new framework.” (page 11, line 312-313 in the Section 4).
6. Page 10, lines 339-340: "According to the independence between Anfinsen’s thermodynamic hypothesis and aggregation". This sentence is not clear: what does "independence" mean in this context?
Response to Reviewer's Comment: This question is similar to the third question we addressed, leading to a similar response from us. Anfinsen’s thermodynamic hypothesis as a tenet of protein folding is strictly limited to the intramolecular conformational changes and relative thermodynamic stability, which are defined by the intramolecular rate (or equilibrium) constants, whereas aggregation is defined by intermolecular rate (or equilibrium) constants. Thus, Anfinsen’s thermodynamic hypothesis is independent of intermolecular aggregation yet interconnected through shared aggregation monomers. Due to this independence, Anfinsen’s thermodynamic hypothesis cannot be challenged by aggregation issues, and vice versa.
7. Page 10, line 351: "rate constants, which operate independently" The "rate constants" cannot "operate". Please rephrase.
Response to Reviewer's Comment: According to reviewer’s comments, we have changed “which operate independently” into “which are mutually independent” (page 12, line 343 in the Section 4).
8. "The reaction Gibbs energy for aggregation reflects deviations from standard concentrations used in standard Gibbs energy calculations, thereby determining whether proteins are metastable or not. "Probably: "Gibbs free energy". Moreover, the sentence is not clear and needs rephrasing.
Response to Reviewer's Comment: Thank you for your feedback regarding the clarity of our description involving reaction Gibbs energy. After careful consideration, we have deleted this sentence from our manuscript to avoid ambiguity and confusion.
Reviewer 2 Report
Comments and Suggestions for Authors
Choi et al. provide a very comprehensive discussion of their model describing the interplay between protein folding and aggregation. The central aspect of their biophysical concept is based on the independent, yet connected, events of intramolecular protein folding and intermolecular aggregation, mediated by "shared aggregation monomers". The present work builds on earlier reports of the authors, especially Refs. 3 and 32, but extends the idea to many more aspects of proteostasis. In particular, this review manuscript addresses aspects such as misfolding vs. aggregation, thermodynamics and Anfinsen's hypothesis, the role of molecular chaperones of different forms (specific and generic ones), stabilizing and destabilizing forces in the folding/aggregation landscape, and strategies to interfere with aggregation in a disease context (proteinopathies).
Consequently, this new paradigm is able to overcome deficiencies of other hypotheses that consider protein aggregation merely as consequence of aberrant protein folding, although protein misfolding may not necessarily lead to aggregation and aggregation may also arise through other pathways independent of protein misfolding.
Many aspects of protein structure and folding are highly relevant for diverse phenomena, both for fundamental biology but also for areas touching upon biomedicine, such as proteinopathies for which the formation of aggregates is a hallmark. Nevertheless, hypotheses and models of protein (mis)folding cannot always explain all observed phenomena. The present manuscript provides interesting viewpoints towards a uniform model of folding and aggregation. Overall, I found the manuscript very enjoyable to read and thought-provoking. It cites literature on the topic extensively and touches upon many different important aspects of protein structural organization. The different sections are supported with figures explaining the key concepts discussed in the related paragraphs.
I have only a few minor points that should be addressed by the authors before publication:
The readability of some figures could be improved. Specifically, the resolution of Figure 1 was not good enough to read smaller text well, and the font size of some labels (e.g., axis labels) of graphs such as those in Figures 2-4 could be increased.
Line 641: "(references)" - Apparently, the references are missing here.
Line 957: "... undergo folding upon folding" should probably read "folding upon binding"
Author Response
We sincerely thank all the reviewers for their insightful and constructive comments, which have been crucial in enhancing both the quality and depth of our manuscript.
To facilitate the review process, in the revised manuscript, all modified or newly added texts are marked in red, and the flanking regions of deleted sections are highlighted in blue for enhanced clarity.
Please note that although page and line numbers of the revised manuscript might change upon uploading to the IJMS system, the color markings will ensure that revisions are easily identifiable.
Reviewer 2
Comments and Suggestions for Authors
Choi et al. provide a very comprehensive discussion of their model describing the interplay between protein folding and aggregation. The central aspect of their biophysical concept is based on the independent, yet connected, events of intramolecular protein folding and intermolecular aggregation, mediated by "shared aggregation monomers". The present work builds on earlier reports of the authors, especially Refs. 3 and 32, but extends the idea to many more aspects of proteostasis. In particular, this review manuscript addresses aspects such as misfolding vs. aggregation, thermodynamics and Anfinsen's hypothesis, the role of molecular chaperones of different forms (specific and generic ones), stabilizing and destabilizing forces in the folding/aggregation landscape, and strategies to interfere with aggregation in a disease context (proteinopathies).
Consequently, this new paradigm is able to overcome deficiencies of other hypotheses that consider protein aggregation merely as consequence of aberrant protein folding, although protein misfolding may not necessarily lead to aggregation and aggregation may also arise through other pathways independent of protein misfolding.
Many aspects of protein structure and folding are highly relevant for diverse phenomena, both for fundamental biology but also for areas touching upon biomedicine, such as proteinopathies for which the formation of aggregates is a hallmark. Nevertheless, hypotheses and models of protein (mis)folding cannot always explain all observed phenomena. The present manuscript provides interesting viewpoints towards a uniform model of folding and aggregation.
Overall, I found the manuscript very enjoyable to read and thought-provoking. It cites literature on the topic extensively and touches upon many different important aspects of protein structural organization. The different sections are supported with figures explaining the key concepts discussed in the related paragraphs.
I have only a few minor points that should be addressed by the authors before publication:
The readability of some figures could be improved. Specifically, the resolution of Figure 1 was not good enough to read smaller text well, and the font size of some labels (e.g., axis labels) of graphs such as those in Figures 2-4 could be increased.
Response to Reviewer's Comment: Following the reviewer's suggestions, we have improved the resolution of Figure 1 and increased the font sizes in Figures 2-4 to enhance readability. These changes are reflected in the revised figures.
Line 641: "(references)" - Apparently, the references are missing here.
Response to Reviewer's Comment: Thank you for pointing out the missing references. We have addressed this oversight by adding three new references (163-165) in the revised manuscript (page 21, line 620 in the Section 7.1.).
Line 957: "... undergo folding upon folding" should probably read "folding upon binding"
Response to Reviewer's Comment: Following the reviewer's suggestion, we have corrected the text from "undergo folding upon folding" to "folding upon binding" in the revised manuscript (page 47, line 1627 in Figure 11 legend). Thank you for bringing this to our attention.
Reviewer 3 Report
Comments and Suggestions for Authors
This is a very interesting manuscript, in which authors discuss the relationships between protein folding and aggregation. The strengths of the work are the inclusion of a large body of experimental data and presentation of a conceptual framework for explaining the results in terms of a common set of intermediates. A wide range of effects is considered, from chaperones to cellular biomolecules and tethering. I recommend publication after authors respond to the comments below.
1. The text is quite long and many arguments are repeated several times. Could these be shortened?
2. It seems evident that all protein properties exhibited in vitro are determined by the sequence, including folding and aggregation, while in the cell other system components may also play a role.
3. The role of pH in folding and aggregation could be discussed as well.
4. In section 4, the Anfinsen hypothesis is discussed. As I see it, this hypotheis states that proteins tend to fold spontaneously to an active structure determined by the sequence. This is equivalent to saying that the folding free energy dG_f < 0 , and says nothing about the magnitude. In a sense this does not cover IDPs. At higher concentrations the aggregation free energy becomes negative for all or most proteins, and they tend to aggregate. The discussion of these effects seems convoluted and over long. Also, the equation given in line 331 works for folding equilibrium only, in presence of aggregation it must include some information about appropriate aggregation equilibria.
5. Analysis of folding and aggregation through attractive and repulsive forces appears somewhat oversimplified. In aqueous solution there are significant entropic effects due to properties of water - the hydrophobic effect and more generally desolvation - that are not directly related to any physical forces, like electrostatic or van der Waals interactions.
Author Response
We sincerely thank all the reviewers for their insightful and constructive comments, which have been crucial in enhancing both the quality and depth of our manuscript.
To facilitate the review process, in the revised manuscript, all modified or newly added texts are marked in red, and the flanking regions of deleted sections are highlighted in blue for enhanced clarity.
Please note that although page and line numbers of the revised manuscript might change upon uploading to the IJMS system, the color markings will ensure that revisions are easily identifiable.
Reviewer 3
Comments and Suggestions for Authors
This is a very interesting manuscript, in which authors discuss the relationships between protein folding and aggregation. The strengths of the work are the inclusion of a large body of experimental data and presentation of a conceptual framework for explaining the results in terms of a common set of intermediates. A wide range of effects is considered, from chaperones to cellular biomolecules and tethering. I recommend publication after authors respond to the comments below.
1. The text is quite long and many arguments are repeated several times. Could these be shortened?
Response to Reviewer's Comment: Thank you for your constructive feedback. We have carefully reviewed the manuscript and made efforts to shorten the text by deleting repeated arguments where possible throughout the manuscript. These deletions have been highlighted by marking the flanking regions of the deleted sections in blue for clarity and ease of review. Given the complexity of the diverse topics discussed under a single unified framework, some repetition may have appeared.
2. It seems evident that all protein properties exhibited in vitro are determined by the sequence, including folding and aggregation, while in the cell other system components may also play a role.
Response to Reviewer's Comment: The reviewer’s insights are crucial and align well with the core themes of our manuscript, highlighting the independent effects of the cellular milieu on protein folding and aggregation. Furthermore, this independence underscores the necessity to redefine the cellular milieu, especially in relation to aggregation. A hallmark yet often overlooked aspect of the cellular environment is the physical linkage of polypeptides of interest with cellular macromolecules. These include interactions with molecular chaperones, macromolecular tethering, and native complexation, which significantly influence protein behavior beyond what is typically observed under simplified in vitro conditions dominated solely by the protein’s intrinsic properties. Thus, the new framework bridges the gaps between the combined protein folding and aggregation in vitro and in vivo. To reflect the reviewer’s comments, we have newly added texts (page 16, line 473-483 and page 17, line 489-495 in the first paragraph of the Section 6).
3. The role of pH in folding and aggregation could be discussed as well.
Response to Reviewer's Comment: Thank you for your valuable suggestion. Indeed, the role of pH in protein folding and aggregation is crucial, as changes in environmental pH can significantly influence the net charges on proteins. Lower or higher net charges at extreme pH can denature proteins by electrostatic repulsions; at the same time, however, these repulsions can inhibit amyloid formation. To reflect the reviewer’s comments, we have newly added texts: “The net charges of proteins are influenced by environmental pH. Proteins denatured at extreme pH can paradoxically resist amyloid fibril formation due to intermolecular electrostatic repulsions (58).” (page 8, line 221-223 in the fourth paragraph of the Section 2).
4. In section 4, the Anfinsen hypothesis is discussed. As I see it, this hypotheis states that proteins tend to fold spontaneously to an active structure determined by the sequence. This is equivalent to saying that the folding free energy dG_f < 0 , and says nothing about the magnitude. In a sense this does not cover IDPs. At higher concentrations the aggregation free energy becomes negative for all or most proteins, and they tend to aggregate. The discussion of these effects seems convoluted and over long. Also, the equation given in line 331 works for folding equilibrium only, in presence of aggregation it must include some information about appropriate aggregation equilibria.
4-1. Response to Reviewer's Comment on the concentration-dependent aggregation free energy: The reviewer’s comment is highly reasonable and insightful. We have revised the text: “In contrast to intramolecular protein folding, where thermodynamics is determined by the N/D ratio that remains unaffected by actual protein concentration, aggregation (or solubility) thermodynamics is significantly influenced by this concentration.” (page 26, line 801-803 in the Section 8.1.). Thus, the traditional comparisons of structural stability between folded (or unfolded) structures and aggregates, based on the 'standard' Gibbs energies for protein folding and aggregation, could be misleading, as these energies do not account for actual protein concentration. The reviewer’s comment aligns with the concept of supersaturation described in our manuscript, where aggregation becomes thermodynamically spontaneous and more stable, once the actual protein concentration—specifically that of shared aggregation monomers—exceeds their solubility limit.
4-2. Response to Reviewer's Comment on “The discussion of these effects seems convoluted and over long”: In response to the reviewer's feedback, we have streamlined the content originally in the fourth paragraph of Section 4 and moved it into the third paragraph of the revised manuscript. This adjustment has effectively clarified the discussion and modestly reduced its length, addressing concerns regarding its previous complexity. The metastability issue needs to be briefly discussed in the Section 4, as it presents a significant challenge to Anfinsen’s thermodynamic hypothesis.
4-3. Response to Reviewer's Comment on appropriate aggregation equilibria: According to the reviewer’s suggestion, we have expanded our discussion of aggregation equilibria by stating " In contrast, the standard Gibbs energy for converting soluble aggregation monomers (D) into fibrils (A) can be given by ΔG_(A-D)^0= -RT ln Ka = RT ln M_s^max, where Ka and M_s^max are the equilibrium constant for elongation and the solubility limit of monomers, respectively (6,7,77)." (page 11, line 301-304 in the Section 4). This equation quantitatively explains how the parameters define the aggregation thermodynamics.
5. Analysis of folding and aggregation through attractive and repulsive forces appears somewhat oversimplified. In aqueous solution there are significant entropic effects due to properties of water - the hydrophobic effect and more generally desolvation - that are not directly related to any physical forces, like electrostatic or van der Waals interactions.
Response to Reviewer's Comment: The reviewer’s point is well-taken. Accordingly, we have revised the text:"Maintaining protein solubility against aggregation is achieved when aggregation-destabilizing forces and factors, such as intermolecular repulsive forces, desolvation penalties, and diverse entropic penalties, energetically outweigh aggregation-stabilizing forces and factors. " (page 21, line 624-627 in the Section 7.1.) In this context, the term ‘factors’ broadly encompasses diverse energetic components, including both indirect physical and entropic influences, that contribute to maintaining protein solubility against aggregation.